# Core–Shell Fibers: Design, Roles, and Controllable Release Strategies in Tissue Engineering and Drug Delivery

**DOI:** 10.3390/polym11122008

**Published:** 2019-12-04

**Authors:** Muhammad Faiq Abdullah, Tamrin Nuge, Andri Andriyana, Bee Chin Ang, Farina Muhamad

**Affiliations:** 1Department of Chemical Engineering, Faculty of Engineering, University of Malaya, Kuala Lumpur 50603, Malaysia; faiq@unimap.edu.my; 2School of Bioprocess Engineering, Universiti Malaysia Perlis, Kompleks Pusat Pengajian Jejawi 3, Arau, Perlis 02600, Malaysia; 3Centre of Advanced Materials, Faculty of Engineering, University of Malaya, Kuala Lumpur 50603, Malaysia; tamrin.nuge@um.edu.my (T.N.); andri.andriyana@um.edu.my (A.A.); 4Department of Mechanical Engineering, Faculty of Engineering, University of Malaya, Kuala Lumpur 50603, Malaysia; 5Department of Biomedical Engineering, Faculty of Engineering, University of Malaya, Kuala Lumpur 50603, Malaysia

**Keywords:** core–shell fibers, scaffolds, tissue engineering, drug delivery, release strategy

## Abstract

The key attributes of core–shell fibers are their ability to preserve bioactivity of incorporated-sensitive biomolecules (such as drug, protein, and growth factor) and subsequently control biomolecule release to the targeted microenvironments to achieve therapeutic effects. Such qualities are highly favorable for tissue engineering and drug delivery, and these features are not able to be offered by monolithic fibers. In this review, we begin with an overview on design requirement of core–shell fibers, followed by the summary of recent preparation methods of core–shell fibers, with focus on electrospinning-based techniques and other newly discovered fabrication approaches. We then highlight the importance and roles of core–shell fibers in tissue engineering and drug delivery, accompanied by thorough discussion on controllable release strategies of the incorporated bioactive molecules from the fibers. Ultimately, we touch on core–shell fibers-related challenges and offer perspectives on their future direction towards clinical applications.

## 1. Introduction

The main goal in tissue engineering is to maintain, enhance and restore various tissue functions. Usually, this goal could be achieved by fabricating a scaffold that closely mimics native extracellular matrix (ECM) [1]. The scaffold should offer desirable architecture and mechanical properties, as well as supporting cell proliferation, differentiation, and migration [1,2,3]. The natural ECM comprises of assorted interwoven protein fibers with size less than hundreds of nanometers [2,4,5]. Therefore, developing nanosize scaffolds that resemble the architecture and features of native ECM is the most challenging area in the tissue engineering field.

The typical scaffold criteria for tissue engineering include biocompatibility, biodegradability, outstanding mechanical properties, and the need for the scaffold to possess highly interconnected porosity with tunable pore size. Nanofibrous tissue scaffolds with these desired criteria can be fabricated through employment of self-assembly, phase separation, and electrospinning, in which electrospinning is the most broadly used technique due to its simplicity, cost-effective and ability to process wide range of polymeric materials [2,6,7,8,9,10]. A highly comprehensive review on electrospinning also has been published just recently, where in-depth overview on electrospinning principle, materials, methods, and applications are provided [6]. In addition, the review also offers perspectives on future development of electrospinning, particularly with regard to scale-up capability, and safety concerns during fiber processing [6]. Although synthetic polymers typically have extraordinary mechanical properties, their application in tissue engineering is deterred by low cell affinity due to lack of cell recognition sites. In most cases, natural polymers, such as collagen [11,12], gelatin [13], silk fibroin (SF) [14], and chitosan [15], are blended with synthetic polymers to improve the biocompatibility of the nanofibers. 

Despite these advancements, the success of scaffold implantation is still low due to inflammation at injury sites caused by implant infection. In addition, certain cytokines are also required to be delivered to the injured area to ensure cell survivability and proliferation. Thus, advanced scaffold design requires further criterion addition; the scaffold must possess the ability to store bioactive molecules (e.g., growth factor, drug, protein, enzyme, gene, etc.), preserve their bioactivity, and control their release in a sustained manner. This is where single or monolithic fibers face their greatest constraint [16]. Biomolecules incorporated into single fibers are typically distributed homogeneously inside the fiber matrix or located in regions near to fiber surface because of phase separation within the fiber. Such environments are problematic for controllable and tunable release, and the bioactivity of biomolecules also could be endangered. 

Core–shell fibers show favorable attributes similar to single fibers, albeit with better prospect to overcome the aforementioned limitations faced by their monolithic fiber counterparts. The initial burst release can be curbed by incorporating bioactive molecules in the core layer, in which shell layer acts as a barrier to dictate the release of the biomolecules. Therefore, appropriate selection of the shell materials is a must, not just to achieve the desired release rate but also to ensure proper cell adhesion of the core–shell fibers. 

In this review, we intend to provide an all-round discussion of core–shell fibers, covering the fibers design, fabrication techniques, roles, and biomolecules release strategies. We begin with an overview of core–shell fibers design, with extra attention was given towards material selection for the core and shell layer. Afterwards, we extensively summarize the recent fabrication methods of core–shell fibers, with focus on electrospinning-based techniques and other emerging fabrication approaches. We then highlight the roles of core–shell fibers in tissue engineering and drug delivery, which, as per our knowledge, has not been discussed in detail previously. We also provide thorough discussion on release strategies of incorporated-biomolecules from core–shell fibers, covering both conventional controlled release and state-of-the-art on-demand release strategies. Ultimately, after a concise summary, we touch on challenges related to core–shell fibers and offer perspectives on their future direction towards clinical applications. As native ECMs in most tissues are constituted of fibrous proteins, only core–shell in fiber form is discussed in this review. Comprehensive review on other core–shell structures, such as core–shell nanoparticles and nanospheres, can be found in the literature [17,18]. While previous literatures [10,16,19,20,21,22] focused only on coaxial and emulsion electrospinning, this review intends to cover recent advances of core–shell fibers regardless of their fabrication methods.

## 2. Designing Core–Shell Fibers with View for Biomedical Applications

A core–shell fiber comprises two separate fiber compartments; the outer compartment (denoted as “shell”), and the inner compartment (denoted as “core”), in which the latter is being completely enclosed by the former. Despite being discrete (often observed by a distinct boundary layer), both core and shell layers are permitting molecular permeation through them. This represents in what is known as diffusion-driven mechanism in biomolecule release studies. In biomedical field, a core–shell fibrous scaffold is designed with the intention to achieve either one or both of these goals: (1) to provide three-dimensional (3D) microenvironment which supports cell culture, and (2) to store and controlling release of drugs and biomolecules for site-targeted delivery or local delivery (commonly referred as “drug depot” or sometimes as “drug highway”). The first goal, or the combination of first and second goals, is closely associated with tissue engineering, whereas the second goal is usually linked to drug delivery application. Figure 1 illustrates two different design approaches of core–shell fibers and their role in delivering multiple bioactive molecules with controllable release profiles for tissue engineering and drug delivery.

The selection of core and shell materials has become very broad, and final materials combination usually takes account the desired physicochemical, biological and mechanical properties of the core–shell fibers, the nature of the bioactive molecules to be loaded, and the release strategies of the biocompounds (Figure 1). Various core–shell material combinations have been reported, including synthetic/synthetic polymer [23,24,25], synthetic/natural polymer [26,27,28], natural/synthetic polymer [29,30,31], and carbon material/synthetic polymer [32,33] combination. In addition, myriad of bioactive molecules also has been incorporated in core–shell fibers, typically in core layer. These include drugs [34,35,36,37], natural extracts [38,39,40], proteins [41,42,43], growth factors (GFs) [12,23,44] and even living cells [45,46,47]. Selected combinations of core–shell fibers materials with or without incorporated biomolecules for various tissue engineering and drug delivery applications are listed in Table 1. The list only includes studies with in vitro cells and/or in vivo animal testing which were reported in the last five years.

A wide range of materials has been employed as core, particularly polymeric-based, materials. Generally, the polymers are categorized as either hydrophilic polymer (natural/synthetic) or hydrophobic polymer (natural/synthetic). In the design of the core–shell fibers, a hydrophobic core was chosen to provide adequate mechanical strength and structural integrity for the fibers, especially if the hydrophilic polymer was intended to be employed as shell layer. However, as the preparation of hydrophobic polymer solution prior to fabrication usually requires use of potentially hazardous organic solvents, future incorporation of bioactive molecules through mixing or blending might prove to be difficult or challenging due to possible loss of molecules’ bioactivity [12,87,88]. As an alternative, hydrophilic material was selected as core layer to provide a “friendly-environment” for the incorporated biomolecules as the hydrophilic solution preparation normally requires the utilization of common and non-toxic solvent such as water and ethanol. Despite this advantage, hydrophilic polymer possesses low mechanical strength and weak structural integrity [23,30]. Thus, the hydrophilic core is typically accompanied with hydrophobic shell in core–shell fibers fabrication. It is also worth noting that in some reported studies, biomolecules can be loaded directly and preserved in the core layer without the presence of the core polymer [12,72,79]. The reason is that the biomolecules are being confined by the spinnable shell layer.

The choice of which shell material to use is, on the other hand, far from straightforward. Shell layer plays a crucial role in core–shell fiber design as favorable cell adhesion and biomolecules release strategy are highly dependent on the selection of shell material. Two general design approaches have been reported as depicted in Figure 1a. In the first approach, hydrophobic polymer was preferred as shell material to take advantage of the slower degradation and/or erosion of the shell layer, hence reducing release rate of drug or bioactive molecules to achieve sustained and prolonged release. However, hydrophobic polymers also have been associated with low affinity towards cells, which will further decrease the applicability of the scaffolds in tissue engineering. Thus, balancing degradation rate and cell affinity is at the highest priority, and this is usually realized through hydrophobic/hydrophilic polymer blends [11,54,89,90] or post-fabrication treatments such as oxygen (O_2_) plasma treatment [30], laser ablation [91], and covalent immobilization of heparin [92]. 

In the second design approach, hydrophilic polymer (usually natural polymer) was favored as shell material. This design approach was chosen to maximize cell–fiber adhesion due to better hydrophilicity and wettability of the shell layer. However, biomolecules loaded into these core-hydrophilic shell fibers typically suffered from burst and faster release, caused by higher degradation and erosion rate of the shell layer. To overcome this, the fibers were usually cross-linked using a cross-linking agent such as glutaraldehyde [34,93]. Although the primary goal of cross-linking is to improve mechanical properties and structural integrity of hydrophilic polymer [39,94,95], it has been reported that cross-linking also enhanced water resistance [95] of the polymer, thus reducing its degradation rate in aqueous solution and eventually slowing down biomolecules release. Nevertheless, several studies have reported the potential toxicity related to glutaraldehyde [94,96]. Hence, novel cross-linking strategies have been proposed in recent years including use of natural cross-linking agents (e.g., citric acid [97], genipin [35,36], and dextran aldehyde or sucrose aldehyde [49]), atmospheric pressure non-equilibrium plasma treatment [94] and ultraviolet (UV) photo-cross-linking [98,99,100] to solve this limitation.

Although polymer selection plays a crucial role in successful biological testing, previous studies show that in vitro cells and in vivo animal testing benefitted the most from the presence and controlled release of bioactive molecules (single, dual, or multiple release) (Figure 1b), in terms of providing biochemical cues for cell signaling or as prevention from inflammation due to infection. For example, human osteosarcoma cells (MG63) and murine 3T3 fibroblasts cells cultured on platelet lyophilisate-loaded core–shell fibers demonstrated higher in vitro metabolic activity compared to those cultivated on free-load fibers [50]. Similar observation was reported in another study by Chen et al. [101]. The aim of their study is to minimize peritendinous adhesion, one of the common complications post-surgery due to tendon injury. The membrane scaffolds incorporated with hyaluronic acid in core and silver nanoparticles in shell layer showed higher synergistic effects in suppressing in vitro attachment and proliferation of adhesion-forming foreskin fibroblasts without exhibiting cytotoxicity, compared to free-load fiber. This was further supported by in vivo study, where nanofiber membranes containing hyaluronic acid and silver nanoparticles demonstrated better peritendinous adhesion prevention than free-load membranes after membrane implantation in New Zealand white rabbit tendons for 3 weeks [101]. 

## 3. Fabrication Techniques of Core–Shell Fibers

Core–shell fibers have been prepared primarily through electrospinning-based approaches including coaxial electrospinning, emulsion electrospinning, and single electrospinning plus in situ or post-treatment. As an alternative to electrospinning-based techniques, other newly discovered or novel fabrication techniques of core–shell fibers also have been reported in recent years. Nevertheless, despite the emergence of other fabrication methods, coaxial and emulsion electrospinning remain as the most widely-used technique to fabricate core–shell fibers. The summary of working principle of different fabrication methods of core–shell fibers is presented below.

### 3.1. Coaxial Electrospinning

The first report of coaxial electrospinning emerged in 2003 [102], where four types of core–shell nanofibers were successfully prepared including sets of different materials (e.g., poly(dodecylthiophene) (PDT)/poly(ethylene oxide) (PEO), polysulfone (PSU)/PEO, palladium (Pd)/poly(lactic acid) (PLA)), and an even pair of identical materials (e.g., PEO/PEO). Since then, coaxial electrospinning has shown tremendous progress in core–shell fibers fabrication experiments (Table 1) and modeling developments particularly due to its reasonably simple setup, low cost, design flexibility and ability to form core–shell nanofibers from broad range of materials [26,39,40,85,90,103]. 

This technique, in principle, uses a special nozzle or needle, which commonly known as coaxial spinneret. This spinneret consists of two nozzles arranged in concentric orientation. Two pumps will then be used to propel two different solutions, to the core and shell nozzle, which lead to the formation of core/shell droplet at the tip of the spinneret. When a high voltage is connected to the spinneret and grounded to a collector, the droplet will be pulled by the electric field and deformed into solution jet. Subsequently, the jet experiences further stretching and thinning due to bending instability. As the fluid jet travels from the spinneret to the grounded collector, the solvent used will be evaporated and this eventually led to the formation of fibers on the collector. 

A number of factors have been shown to significantly influence surface morphology, diameter, mechanical properties, porosity, and pore size distribution of the coaxial electrospun materials. These factors can be categorized as electrospinning, solution, and environmental parameters [104]. Electrospinning parameters include applied voltage, distance from needle to collector, and core–shell flow rate ratio. Although solution parameters include polymer concentration, viscosity, and conductivity, temperature and relative humidity are two factors that been classified as environmental parameters. Detailed discussion on the effects of these parameters in conventional coaxial electrospinning is not provided, as it has been comprehensively reviewed elsewhere [19,20,105]. The focus discussion of this section is shifted to advanced and complex coaxial electrospinning instead, with thorough discussion is presented in the next paragraphs.

Although conventional coaxial electrospinning has served its purpose, recent developments of advanced and complex coaxial electrospinning also have been reported. Song and co-workers have utilized magnetic field-assisted electrospinning to fabricate highly aligned laminin-polydioxanone (PDO)/collagen core–shell nanofibrous matrices for potential use as biofunctional scaffolds that promote neuritogenesis [61]. In this advanced electrospinning technique, external magnetic field is introduced at the collector by arranging the grounding and insulating regions alternately. When the electrical and magnetic fields are supplied, the electrostatically charged polymer fibers are stacked uniaxially on the collector, yielding highly aligned nanofibrous matrices. In this study, interconnected pore structures which are similar to natural ECM have been observed. Further biological testing with HT-22 hippocampal neuronal cells revealed that surface hydrophilicity is strongly affecting initial cell adhesion, whereas cell proliferation is influenced by surface hydrophilicity and the presence of collagen and laminin [61]. Furthermore, the highly aligned laminin-PDO/collagen nanofiber has also been shown to successfully guide neurite outgrowth and stimulate neurogenic differentiation through aligned topography and controllable release of laminin [61].

In another work, micropatterned fiber matrices were fabricated by deposition of coaxial electrospun fibers onto lithographic collectors to mimic myocardium anisotropic structure [32]. Prior coaxial electrospinning, three photomask patterns, including square (Sq-), rectangle (Rect-) and honeycomb (Hc-) patterns, were designed and printed using electron-beam mask lithography system as depicted in Figure 2a. The printed photomask was laid onto a glass substrate, followed by exposure to lithograph machine to obtain micropatterned circuits. These circuits were then used as collector during coaxial electrospinning. Carbon nanotubes (CNTs)-poly(ethylene glycol)-poly(DL-lactide) (PELA) core–shell Hc-patterned scaffolds exhibited higher strain to failure and ultimate tensile strength, as well as more spacious microenvironment compared to Sq- and Rect-patterns. In addition, Hc-patterned scaffolds showed higher cell viabilities and more formations of capillary-like networks during coculture of primary cardiomyocytes (CMs) with endothelial cells (ECs) and cardiac fibroblasts (CFs) (Figure 2b,c) [32].

Combinations of coaxial electrospinning and electrospraying also have been reported, with an aim to introduce micro- or nanoparticles onto the surface of core–shell nanofibers. Birajdar and Lee [106] combined electrospraying and coaxial electrospinning in parallel, when proposing novel uncorking strategy for smart control release of model drug from core–shell nanofibers. In their study, silica nanoparticles were electrosprayed on core–shell nanofibers, and they function as corks on the fiber surface. Upon sonication, the corks are triggered and began to detach, which eventually lead to the release of drug from core–shell fibers. Meanwhile, similar fabrication technique also was employed by Aragón et al. [107] to functionalize poly(ɛ-caprolactone) (PCL)-nanohydroxyapatite (nHAP)/PCL or PCL-nHAP/poly(vinyl acetate) (PVAc) core–shell fibers with bone morphogenetic protein-2 (BMP-2) loaded-poly(d,l-lactic–*co*–glycolic acid) (PLGA) microparticles. They revealed that the structure and size of core–shell fibers and microparticles were unchanged after the PLGA particles were electrosprayed. In addition, the presence and controlled release of BMP-2 from core–shell fibrous mats was shown to improve proliferation and viability of human osteoblasts, as well as stimulating gene expression of osteoblasts maturation markers and bone formation [107].

On the other hand, more complex coaxial electrospinning has been reported in form of triaxial and needleless electrospinning. Triaxial electrospinning corresponds to the use of triaxial spinnerets arranged in concentric position. However, it may or may not lead to the formation of trilayer fibers. In most reported triaxial studies, the outer working fluids are usually a solvent or mix of solvents without the presence of polymer [108,109,110,111]. In this case, the solvents were introduced in the outer layer to improve electrospinnability of polymers in intermediate layer which will increase jet stability during electrospinning or preventing clogging when highly volatile solvents are used.

To give an example, a novel drug-protein nanocomposite encapsulated by cellulose acetate (CA) had been produced by employing modified triaxial electrospinning as portrayed in Figure 3a–c [110]. In this fabrication process, spinnable ibuprofen was used as core solution, whereas the unspinnable CA was used as intermediate fluid. To enhance the electrospinnability of CA, a solvent mixture of acetone/acetic acid was used as the shell working fluid. It has been reported that the thickness of CA layer could be precisely tuned via adjustment of CA concentration in the intermediate solution. Further characterization using scanning electron microscopy (SEM) and transmission electron microscopy (TEM) revealed that the fibers have linear and cylindrical topography with clear core–shell structure. The CA layer helped in prolonging the release duration of ibuprofen, in which thicker layer led to more sustained and longer release period [110]. The formation of trilayer nanofiber through triaxial electrospinning also had been proved to be not impossible [112,113], although current technique is restricted by the need to use the core, intermediate and shell solutions with similar physicochemical properties.

Despite its importance and recent progress, coaxial electrospinning is commonly associated with very low production rate. Consequently, alternatives to coaxial electrospinning with better production capacity have been proposed. One of the alternatives is through the employment of needleless coaxial electrospinning technique through usage of a simple spinneret setup (Figure 3d,e), which can potentially exceed the production capacity of conventional coaxial approach [95]. The proposed weir spinneret enables the solution to be electrospun from free liquid surface. It has been reported that this needleless approach led to formation of core–shell fibers with better core–shell uniformity and ratio [95]. This might be helpful in fine tuning of the degradation rate, which is desirable for drug release application.

### 3.2. Emulsion Electrospinning

Emulsion electrospinning is an outstanding alternative to coaxial electrospinning for the fabrication of core–shell fibers. It is relatively simpler than coaxial electrospinning in term of equipment setup and the fact that it only requires the use of conventional single needle for electrospinning. A water-in-oil (W/O) emulsion was prepared by dispersing biomolecule-containing aqueous solution (water phase or dispersed phase) droplets in a mixture of polymer solution (oil phase or continuous phase) and a surfactant (e.g., Span 80, Tween 80, Pluronic F-68, etc.), followed by vigorous stirring overnight or sonication to obtain uniform W/O emulsion. An oil-in-water (O/W) emulsion also can be prepared vice versa, by suspending oil droplets (dispersed phase) in water phase solution (continuous phase). The selection of emulsion system (W/O or O/W) generally depends on classification of the intended incorporated bioactive molecules (hydrophobic or hydrophilic). As a result, W/O emulsion was employed to disperse hydrophilic biomolecules, whereas O/W emulsion was used to suspend hydrophobic molecules or drugs. Subsequently, the emulsion was drawn into a syringe and subjected to single electrospinning.

Surfactant plays an important role in emulsion electrospinning. Surfactant contains a hydrophilic head which directs toward polar or aqueous phase and a hydrophobic tail (hydrocarbon or fluorocarbon) which points toward oil phase [114]. The main role of a surfactant is to improve the stability of an emulsion by reducing interfacial tension between water and oil phases [114,115]. Proper selection of surfactant and its concentration is also vital to obtain desired colloidal/droplets size (macroemulsion, microemulsion, or nanoemulsion) and appropriate electrical charge of the particles [19,115,116], all which are critical factors that influence the quality of the as-produced core–shell fibers.

The formation of core–shell structure during emulsion electrospinning is initiated by rapid evaporation of oil phase (continuous phase). This will lead to the increase of oil phase viscosity, which prompts the biomolecule-containing aqueous droplets to migrate towards the center of the solution jet. Under the influence of electric field, the jet is stretched towards the grounded collector, and core–shell fibers are collected when all of the solvents are evaporated. Conventional emulsion electrospinning has been studied extensively, and various bioactive molecules, including drug [72,117,118], GF [44,119], protein [43], and antioxidant [120], have been incorporated in core–shell fibers via this method. In addition, three-dimensional (3D) microfibrous scaffold with core–shell architecture for potential use as regenerative skin tissue also has been reported to be prepared by emulsion electrospinning [121].

The reports on more advanced and complex emulsion electrospinning also have emerged in recent years. For instance, emulsion electrospinning has been combined with in situ electrospraying to produce three-compartment drug delivery system with three different release profiles [37]. Two drug compartments are created from core–shell structure while additional compartment (microspheres) is introduced through electrospraying. As a result, three different drugs can be loaded in this system, each in core (ampicillin), shell (Bay 11-7082), and in electrosprayed microspheres (pirfenidone) [37]. Although it is not discussed in detail in the report, the drug release rate can be controlled by adjusting the processing parameters of emulsion electrospinning and electrospraying. In another study, bicomponent core–shell scaffolds have been prepared by utilizing dual-source emulsion electrospinning [59]. The aim of the system is to provide dual delivery of GFs, and this was achieved by encapsulating nerve growth factor (NGF) and glial cell line-derived neurotrophic factor (GDNF) individually in poly(d,l-lactic acid) (PDLLA) and PLGA fibers, respectively by emulsion electrospinning. A current study of needleless emulsion electrospinning also has been reported, which aims to overcome low productivity of conventional emulsion electrospinning [122]. Horseradish peroxidise (HRP) and multiple GFs (transforming growth factor-β (TGF-β), basic fibroblast growth factor (bFGF), and insulin growth factor-1 (IGF-1)) have been successfully incorporated in the core layer of this system. In vitro testing revealed that the as-prepared core–shell fibers promote viability and metabolic activity of porcine mesenchymal stem cells (MSCs) [122].

### 3.3. Single Electrospinning Plus In Situ or Post-Treatment

Although the fabrication approaches of core–shell fibers have been dominated by coaxial and emulsion electrospinning, another viable electrospinning-based fabrication strategy for core–shell fibers has been reported in form of single electrospinning plus in situ or post-fabrication treatment. The first method employing this strategy is single electrospinning plus in situ phase separation and subsequent crystallization [99,123,124,125,126]. Phase separation favors only binary polymers which are incompatible with each other, and they must be able to be dissolved in the same solvent system. Otherwise, phase separation will not occur, and subsequently no core–shell structure forms. For instance, Xu et al. [125] reported the preparation of PLA/chitosan core–shell fibers via this method (Figure 4a). Chitosan was added to improve cell biocompatibility of the PLA fibers, as well as potentially influence the topography of the fibers. Both PLA and chitosan were dissolved in trifluoroacetic acid (TFA) at various PLA/chitosan weight ratio of 90:10 to 10:90 before they were subjected to single electrospinning at 25 °C. In an almost similar principle to emulsion electrospinning, phase separation was observed when the solvent evaporated during electrospinning. As a consequence, PLA/chitosan solution slowly turned to be metastable or unstable which led to spinodal decomposition or nucleation and growth (phase separation) [125]. The cationic chitosan was preferred to migrate out towards the outer layer of the solution jets under electrostatic repulsion, hence forming the shell layer of the core–shell fibers.

This research group also revealed another notable finding: the core–shell structure can be switched to “island-like” topography simply by conducting electrospinning at higher temperature (35, 45, 50, and 60 °C) (Figure 4a). The appearance of island structure is an interesting phenomenon, and its formation mechanism was speculated to be due to quicker solvent evaporation at higher temperature, thus some molecular chains of chitosan were unable to migrate towards the jets surface. As a result, there was inadequate chitosan component to surround the whole fiber surface, hence the reason on formation of island structure instead of core–shell structure [125]. The island-like structure also was shown to promote better spreading of mouse preosteoblasts MC3T3-E1 cells than core–shell topography (Figure 4b) [125]. As cell adhesion was strongly influenced by fiber surface properties, the moderate hydrophilic interfaces and rougher surface topology of the island-like structure provide a better platform for preosteoblasts attachment and spreading than core–shell architecture.

Alternatively, the preparation of core–shell fibers can also be realized through combination of single electrospinning and post-fabrication treatments such as UV photocrosslinking [68], metal sputtering, and electrochemical deposition [127], as well as reoxidation with thermal pretreatment [128]. Wang et al. [68] reported the preparation of core–shell-based fibrous and sheet composite scaffolds by encapsulating aligned nanofiber yarn (NFY) inside a UV photocurable hydrogel shell as portrayed in Figure 5. The aligned NFYs were comprised of PCL, SF, and polyaniline (PANI), and they were fabricated via dry–wet single electrospinning method (collected in water/ethanol bath before they were rolled using a rotating cylinder). For the subsequent preparation of core–shell fiber, a single NFY was encapsulated with poly(ethylene glycol)–*co*–poly(glycerol sebacate) (PEGS-M) in a poly-(dimethylsiloxane) (PDMS) microtube. A photoinitiator (Irgacure 2959) was added to PEGS-M solution initially, and photopolymerization of PEGS-M was achieved after exposure to UV irradiation (365 nm) at ~12 mW/cm^2^ for 30 s [68]. Similar photopolymerization procedure was performed to prepare mono- and bilayer core–shell sheets, which were later found to promote myogenic differentiation and induce formation of elongated myotube [68], demonstrating an ideal platform for skeletal muscle regeneration.

In another study, Beregoi et al. [127] proposed a novel synthesis route for polyaniline (PANI)-coated fiber webs through combination of single electrospinning and successive electrochemical deposition of PANI. Prior PANI deposition, poly(methyl methacrylate) (PMMA) electrospuns were gold-coated, and they function as working electrodes during electrochemical deposition of PANI. These core–shell fiber webs showed good biocompatibility against eukaryotic cells and they demonstrated exciting electrochromic properties, which is the ability to reversibly change their color when applied voltage was switched unceasingly from 0 to 1 V, highlighting their potential as smart artificial muscles [127].

Another interesting synthesis route of core–shell fibers also have been reported recently by Zhang’s group where inorganic core–shell nanofibers were successfully constructed via single electrospinning followed by reoxidation process with thermal pretreatment [128]. Precursor electrospinning solution was prepared by mixing poly(vinyl alcohol) (PVA), NaYF_4_:Yb^3+^,Er^3+^ microrods solution, and silica gel in deionized (DI) water. This solution was subjected to single electrospinning to yield precursor nanofibers with diameter ranging from 450 to 550 nm. The precursor nanofibers were then pretreated at minimum temperature of 200 °C, followed by reoxidation treatment using an oxidation solution. The combination of thermal pretreatment and reoxidation is necessary to prevent fiber cracks and reinforce the silica shell, which provides sufficient strength for the nanofibers. Interestingly, the microrods are embedded in an end to end arrangement in the fiber along fiber direction to yield core–shell structure. This phenomenon was hypothesized to be due to electrostatic drawing during electrospinning [128]. The core–shell nanofibers were later incorporated with two different drugs (ibuprofen and doxorubicin) and evaluated for simultaneous monitoring of dual drug delivery.

### 3.4. Other Fabrication Techniques

In addition to electrospinning-based strategies, other state-of-the-art fabrication techniques for core–shell fibers also have been reported recently. These include coaxial electrohydrodynamic (CEHD) direct-printing [55,129], self-assembly [130,131,132], coaxial bioprinting [45], emulsion centrifugal spinning [50], solution blow spinning [73,133], coaxial airbrush [89], and microfluidics [134,135,136,137,138]. CEHD direct-printing has been proposed as a viable alternative for coaxial electrospinning, in which this technique provides better control of the interstitial porosity of the as-produced fiber compared to coaxial electrospuns. Greater control of the fiber deposition can be achieved through pre-programmable movement of either the collector or the coaxial spinneret itself. For example, Yao et al. [129] produced tetracycline hydrochloride-loaded polyvinyl pyrrolidone (PVP)/PCL core–shell fibers via this CEHD method. The collector has been preprogrammed to move in designated X-Y pattern. During electrohydrodynamic (EHD) process, well-ordered fibers were obtained, and the process can be repeated to yield 3D multiple-stacked fiber layers for potential application as drug patches. Nevertheless, the diameter of the as-produced fibers was still relatively large (~30 µm) due to shorter working distance (2 mm) employed in this technique [129]. The optimization of process and solution parameters is well underway, with the aim to reduce fiber diameter size to reach within nanometer range.

Another fascinating and powerful strategy to produce core–shell fibers is through self-assembly. The principle of the technique is based on the ability of a structure to construct and organize itself from a collection of disordered precursor blocks [131,132]. An example of core–shell nanofibers production via self-assembly has been reported by Wang’s group (Figure 6) [131]. In their study, molecularly solubilized di-block copolymer poly(ethylene glycol)–*b*–poly(4-vinylpyridine) (PEG–*b*–P4VP) was used as precursor block for the designated self-assembly. Below critical water content (CWC), PEG–*b*–P4VP interacts with deoxyribonucleic acid (DNA) chain to form linear DNA/polymer complexes where each DNA chain was surrounded by P4VP chains. Upon gradual increase of water content, P4VP block chains were allowed to aggregate which eventually induces change conformation of the DNA chain, prompting it to surround the P4VP aggregates [131]. Ultimately, this led to the formation of monodisperse core–shell nanofibers. Self-assembly is attractive in a way that it avoids the need for the complex equipment setup. However, its complicated formation mechanism may prove challenging for its widespread uses.

Despite its prevalent use, one of the main criticisms of coaxial electrospinning is its lack of ability to yield 3D core–shell fibrous scaffolds. Coaxial bioprinting has been reported as a promising alternative to fabricate core–shell fibers in 3D arrangement. In a work reported by Mistry et al. [45], three different cell-types (HepG2 cells, 3T3 mouse fibroblast (3T3) cells, and primary human umbilical vein endothelial cells (HUVECs)) have been incorporated separately in core–shell laden strands through coaxial bioprinting method (Figure 7). In this study, respective cell type was combined with core hydrogel which consists of type I rat tail tendon collagen, Matrigel, and methacrylated gelatin (GelMA), although poly(ethylene glycol) diacrylate (PEGDA), alginate, or PEGDA-alginate blend is used as shell gel [45]. The core–shell strands were then printed using a coaxial spinneret mounted on a commercial 3D bioprinter. The authors also reported successful incorporation of dual cells; HUVECs in the core and HepG2 cells in the shell layer of the 3D strands [45]. Currently, bioprinting technique is attracting ever-increasing attention as it allows formation of complex 3D geometries which is highly favorable for potential application as tissue scaffolds. However, despite its advancements, several bioprinting-related challenges are yet to be resolved. These include short shelf-life and storage issue of the bioink, slow printing process, and restricted mechanical strength of 3D scaffolds [139].

Meanwhile, Buzgo et al. [50] developed another novel technique to prepare protein-embedded core–shell fibers through centrifugal spinning method. This technique was proposed as the solution to poor cell penetration which is commonly associated with electrospun fibers (caused by limited layer thickness and small pore size). Initially, a W/O emulsion was prepared by dropping platelet lyophylisates protein/Pluronic F-68 water phase into PCL oil phase, followed by sonication. The emulsion solution was then subjected to centrifugal spinning process at rotation speed of 11,000 rpm (Figure 8). This technique is based on force spinning technology, in which high centrifugal forces are generated during fast spinning, resulting in formation of fibers [50]. Similar to emulsion electrospinning, fast evaporation of oil phase solvent increases oil phase viscosity, which in turn forces protein-containing water phase to migrate towards the center of the fiber, hence producing core–shell structure. Although this method can produce core–shell fibers in nano-size in large volume, future development of aligned or patterned fibers might be difficult and challenging. In the meantime, the working principle, advantage, and limitation of other fabrication strategies of core–shell fibers are summarized in Table 2 below.

## 4. Roles of Core–Shell Fibrous Scaffolds in Tissue Engineering and Drug Delivery

Core–shell fibers have attracted enormous attention from the scientific community, not just for potential applications in tissue engineering and drug delivery fields, but also for non-biomedical applications such as sensors [140], lithium (Li)-ion batteries [141], supercapacitor [142], membranes lighting [143], carbon dioxide (CO_2_) adsorbents [144], and desalination [145], to name a few. However, most of core–shell fibers applications have been directed towards tissue engineering and drug delivery due to their unique and novel attributes that ideally suits these applications. The roles and importance of core–shell fibers in tissue engineering and drug delivery are highlighted in this section.

### 4.1. Form Fibers from Almost Any Material

Although electrospinning is simple yet powerful technique to fabricate nanofibers, not every polymeric material is readily spinnable. Some of emerging novel materials which show superior mechanical or biological properties such as poly(glycerol sebacate) (PGS) [34,54,92], acrylated poly-(l-lactide–*co*–trimethylene carbonate) (aPLA–*co*–TMC) [100] and lecithin [108], as well as some natural polymers (e.g., collagen [31], SF [29,146], zein [38], and levan [69]) are reported to possess poor electrospinnability. Core–shell structure provides solution to this impasse by allowing fiber formation of these materials. Four different approaches have been implemented: (1) employing unspinnable material as core layer and spinnable material as shell layer (also known as “traditional coaxial electrospinning”), (2) employing spinnable material in core and unspinnable material as shell layer (referred as “modified coaxial electrospinning”), (3) employing unspinnable material as core layer and use solvent as shell fluid, and (4) blending unspinnable material with another material with superior spinnability, followed by phase separation.

In the first approach, core–shell structure is used to confine the unspinnable material in the core layer, followed by the removal of the shell layer to produce single fiber of the initially unspinnable material. PGS for instance, is rapidly attracting attention as one of the promising materials for tissue engineering scaffolds. However, PGS suffers from deprived electrospinnability due to its low molecular weight and lack of chain entanglement [34,54,60,92]. In most cases, PGS beads instead of fibers are formed when the solution is being electrospun. To overcome this limitation, You and co-workers successfully fabricated the typically poor electrospinnability PGS fibrous scaffolds using PLA as the shell material [54]. PLA shell layer was subsequently washed away using chloroform, followed by curing in vacuum oven at 120 °C for 24–48 h to obtain single PGS fibers. 

Similar procedure has been reported in another study for the formation of anisotropic collagen. Previously, collagen hydrogel is widely used in tissue engineering application. However, the hydrogel form has a homogeneous and isotropic structure, which is dissimilar to the anisotropic structure of the native tissues. To obtain anisotropic collagen, collagen/PVP core–shell nanofibers were fabricated by coaxial electrospinning, followed by subsequent removal of PVP using ethanol [31]. The as-produced anisotropic collagen was shown to promote growth of HUVECs and direct their orientation along fiber direction [31]. Meanwhile, the shell layer is not necessarily needed to be removed post-fabrication. Tian et al. [29] used PLA that remained in shell layer to confine the unspinnable pure SF in the core, provide satisfactory surface hydrophilicity, and appropriate biological cues for the growth and differentiation of PC12 cells.

The second approach contradicts the first one, where in this approach the material with good spinnability is employed as the core layer. This material acts as a fiber template which allowed unspinnable materials to surround it and eventually formed shell layer. For example, levan has found its application in tissue engineering and drug carrier due to its heparin mimicking activity and decent biocompatibility [69,147,148]. Nevertheless, it was found to have insufficient entanglement and inferior molecular flexibility which makes it harder to form homogeneous fiber. Implementing the second approach, Avsar et al. [69] used spinnable PCL and PEO as core materials, and levan was employed as shell polymer. PCL/levan and PEO/levan core–shell fiber scaffolds were successfully obtained, and they demonstrated promising results in reducing neointimal proliferation and thrombogenicity of prosthesis and grafts [69], which are favorable for cardiac tissue engineering.

The third approach is specifically employed to allow formation of fiber from materials with high viscosities. Single electrospinning trials of these materials usually resulted in clog formation and needle blockage. Zein emerged as outstanding and promising material for tissue engineering applications [38,149]. However, forming zinc fibers via single electrospinning is problematic due to its high viscosity [38]. As a solution, aqueous ethanol was deployed as the shell fluid to ensure the continuity of the electrospinning process. Different ethanol concentrations also were shown to influence fibers surface hydrophobicity and topography [38]. In another study by Yang et al. [108], anhydrous ethanol was used as shell solution in triaxial electrospinning to prevent clogging and to enhance the uniformity of the as-prepared fibers. In this study, unspinnable lecithin was used as core material and spinnable Eudragit S100 as intermediate material. This represents the combination of the first and third approach. 

In the fourth approach, the spinnability of the unspinnable material is improved simply by blending it with spinnable material. However, to yield core–shell fiber, both the unspinnable and spinnable materials must be incompatible, so that discrete core and shell layers can be obtained through subsequent phase separation. aPLA–*co*–TMC is a novel block copolymer with outstanding mechanical properties and low toxicity of its degradation products [100]. These key attributes make it as one of the promising materials for vessel tissue engineering especially for in vivo implantation. Nevertheless, aPLA–*co*–TMC solution has low molecular weight which makes it difficult to process and form homogeneous fiber. Stefani and Cooper-White [100] added spinnable PCL to aPLA–*co*–TMC solution to improve its spinnability. The incompatibility of PCL and aPLA–*co*–TMC melts allowed the formation of novel core–shell fibers through single electrospinning coupled with in situ rapid UV cross-linking of phase-separated aPLA–*co*–TMC shell. PCL/aPLA–*co*–TMC core–shell fibrous scaffolds were shown to promote adhesion, proliferation, and alignment of human mesenchymal stem cells (hMSCs) in vitro, as well as demonstrating mechanical characteristics similar to native human arterial tissues [100].

### 4.2. Modify Physical and Mechanical Properties of Fibers

Physical and mechanical properties improvement of monolithic fibrous scaffolds can be realized simply by blending two or more different polymers prior fiber fabrication. However, this approach is restricted as both polymers are required to be miscible with each other, and a single solvent system that is able to dissolve both polymers is needed to be used. This will significantly limit the range of polymers which can be used in scaffolds preparation. Core–shell fibers offer unique opportunity to overcome this limitation by allowing modification as well as enhancement of these physical and mechanical properties regardless of polymers miscibility. 

The use of natural polymers in scaffolds preparation is crucial to support cell attachment and proliferation. Among the extensively explored natural polymers for tissue engineering scaffolds are gelatin [27,34,36,70,94,150], collagen [28,53,55], chitosan [26,125,151], and CA [62,123,152,153], to name a few. Despite offering a favorable cell-friendly interaction, most natural polymers are susceptible to fast degradation, and they possess relatively poor mechanical properties. For instance, gelatin has showed remarkable potential as substitute for native skin. However, the mechanical properties of gelatin-based engineered skin are mismatched to those of native skin, with the biggest discrepancy observed in vitro and in the early stage of the scaffold engraftment. This mechanical difference will eventually lead to difficulty during grafting, vulnerability to shear post engraftment, and reduced elasticity and strength. To overcome this, pure PLA and PCL are incorporated as core layer to enhance biomechanical properties of gelatin-based engineered skins [27]. The mechanical properties of core–shell engineered skins were then compared to monolayer gelatin scaffolds. In vitro mechanical properties of core–shell engineered skins were significantly improved compared to single gelatin fibers. Although the mechanical advantage of core–shell grafts was lost after engraftment to full-thickness excisional wounds in athymic mice, increased inflammatory response to core–shell engineered skins was detected, with significant presence of M2 macrophage and increased upregulation of interleukin-6 (IL-6) expression [27]. Further investigation of the inflammatory response to core materials is needed to be conducted to optimize this approach for clinical application. 

Additionally to its deprived mechanical strength, gelatin also suffers from poor water resistance. In most cases, gelatin is required to be cross-linked to avoid fast solubilization in aqueous medium. However, the most common cross-linking agents are associated with risks of toxicity, thus making them less desirable for biomedical applications. Liguori and co-workers have synthesized genipin/gelatin core–shell electrospun nanofibers and utilized alternative cross-linking approach; namely pressure plasma treatment to decrease the deformation at break of the gelatin layer, as well as improved structural and morphological stability of the fibers during soaking in aqueous solution [94].

Similar to gelatin, CA fibers also suffer from poor mechanical performance and fast degradation which limits its applicability in biomedical field [62,153]. Core–shell fibers design provides solution to this constraint by allowing the incorporation of another polymer with supreme mechanical properties in the core layer to improve the overall mechanical performance of the fibers. Hua et al. [153] proposed polyurethane (PU)/CA core–shell fibers design with superior mechanical properties than the previously reported pure CA fibers. The incorporation of 28% PU in core layer of PU/CA core–shell fibers resulted in a 60-fold increase in tensile strength when compared to pure CA fibers. The incorporated PU also was shown to significantly improve elongation at break to about 14% compared to 4% of pure CA fibers. The PU/CA core–shell fibers were evaluated as semen sensitive delivery system, and in vitro release study revealed that the fibers remain unbroken in simulated vaginal fluid (SVF) at pH 4.2 whilst rapid dissolution was observed during in vitro study in phosphate buffer saline (PBS) at pH 7.4 [153]. Both mechanical and in vitro release studies demonstrated that PU/CA core–shell fibers are highly promising for pH responsive delivery, especially for intravaginal drug delivery.

Synthetic polymers, on the other hand, generally possess superior mechanical strength than natural polymers. PCL is one of the most studied synthetic polymers for tissue engineering. Despite its outstanding mechanical qualities, the hydrophobicity of PCL and its limited cells recognition-sites has further hampered its applicability as graft substitutes. Duan et al. [28] overcame this issue by employing collagen as shell material in coaxial electrospinning set-up, followed by cross-linking with genipin and heparin, which led to improvements in tensile strength, stitching strength, bursting pressure, swelling ratio, and decomposition temperature. The PCL/collagen core–shell electrospun nanofibers also exhibited good biocompatibility and cell affinity towards ECs and smooth muscle cells (SMCs) during cytotoxicity and cell infiltration tests [28]. 

Similar design approach of PCL/collagen combination also has been reported when multilayered PCL/collagen core–shell ultra-fine fibrous struts were prepared by combination of EHD jet and bioprinting process [55]. Highly porous and mechanically controlled fibrous scaffolds with high in vitro cells activity (mouse preosteoblast (MC3T3-E1) cells) were prepared in this study by optimizing processing condition and employing simple coating using type I collagen [55]. Apart from collagen, several other polymers such as chitosan [26], GelMA [98], gelatin [34,36,150], and PVAc [91], have been employed as shell material to improve the hydrophilicity and cell affinity of PCL.

PLA also has been the subject of extensive studies in biomedical field in recent years due to its inherent biocompatibility and biodegradability. However, the hydrophobic nature and poor mechanical strength of PLA hamper its applicability in biomedical application, especially for tissue engineering. PVA was used to overcome this limitation, in which the utilization of PVA enhanced the hydrophilicity of pure PLA and improved its overall tensile strength up to 254% [25]. The PVA/PLA core–shell nanofibrous scaffolds also were shown to stimulate cell growth and attachment of human embryonic kidney (HEK) cells (HEK-293). 

Meanwhile, PGS was designed for soft tissue engineering because of its exceptional ability to recover from deformation. Nevertheless, its application was limited by its low viscosity, making fabrication difficult in fiber form. Therefore, an additional physical or chemical process is required to cure PGS into solid fibers. In a work by Hou et al. [92], PCL and heparin were incorporated as shell layer and anticoagulation agent, respectively to improve physical and biological properties of PGS/PCL core–shell fibers for tissue engineering application. The slowly degraded PCL layer provides mechanical support and structural integrity of the fibers. The as-produced PGS/PCL core–shell nanofibrous scaffolds showed improvement of Young’s modulus from 5.56 to 15.7 MPa, ultimate tensile stress from 2.04 to 2.91 MPa, and elongation capacity from 291 to 907% as compared to monolithic PCL fibers. The incorporated PGS and heparin also were observed to stimulate the attachment and proliferation of HUVECs [92].

Mechanical strength of scaffolds is one of the critical criteria which will determine the suitability of the scaffolds for tissue engineering applications. Although enhancement of the mechanical properties of the core–shell fibers was clearly observed from the experimental results, the underlying mechanism, which leads to the mechanical improvement, is still not conclusively defined. A number of explanation were offered in the previous studies and in general, the mechanical properties of core–shell fibers are reported to be influenced by several factors including fiber diameter and morphology [92,154], overall fiber mat porosity or fiber packing density [25,154,155], volume or weight fractions of core and shell components [55,57], physical and/or chemical interactions between core and shell layer [25,156] and the molecular alignment in the polymer chains [49]. For example, as described in the previous paragraph, Hou and co-workers reported a better mechanical performance of PGS/PCL core–shell nanofiber scaffolds compared to pristine PCL nanofiber mats [92]. During fabrication of PGS/PCL core–shell nanofibers, core (*i.e.*, PGS) ratio was increased which resulted in fiber diameter increases. They speculated that larger fiber diameter of core–shell nanofibers leads to higher Young’s modulus and tensile strength values, while the increased ratio of PGS which is ductile by nature also resulted in superior elongation capacity of the core–shell nanofibrous scaffolds [92].

Meanwhile, Horner et al. [154] revealed that the tensile moduli of PEKK/PCL and gelatin/PCL core–shell nanofibrous scaffolds was influenced by both bulk scaffolds’ porosity and mechanical properties of individual fibers. In term of individual fiber mechanics, higher tensile modulus of core–shell scaffolds was observed as a result of their larger fiber diameter, caused by the increased of volume fraction of the core components. Bulk scaffolds porosity also plays a pivotal role in influencing the resultant mechanical properties of the core–shell scaffolds after certain porosity threshold (in this case, ~85% porosity for PEKK/PCL and gelatin/PCL systems) [154]. From their findings, it is demonstrated that larger fiber diameter also resulted in increased porosity. Below porosity threshold of ~85%, the tensile modulus was increased although the porosity is increasing: the tensile modulus was speculated to be more significantly influenced by the effects of individual fiber mechanics (larger fiber diameter). However, the effects of bulk scaffold porosity being dominating at above 85% porosity, and as a result lower tensile modulus was obtained. The reason is that at higher porosity, fiber packing density is low, and during tensile loading which is parallel to fiber direction, the fibers try to align longitudinally as a response to the applied force. Presumably, the weight of fibers at a certain dimension is low due to low fiber packing density and this resulted in poorer tensile performance [154]. This is in agreement with the mechanism reported in other studies [25,155], which implied that higher fiber packing density corresponds to superior tensile modulus. 

It is also worth noticing that the mechanical properties of core–shell fibers can be tailored accordingly by simply adjusting the weight or volume fractions of the core components and the types of core polymer used. The mechanical properties of the core–shell systems can then be predicted by the classical and simple rule of mixtures [157,158]. To give an example, increasing volume fraction of the stiffer PLLA in the core layer of PLLA/PLCL core–shell system resulted in improved elastic moduli of the core–shell fibers by about 145 times, as compared to pure PLCL fibers [57]. Meanwhile, the mechanical performance of core–shell fibers also critically depends on the physical and chemical interactions between core and shell layer. Jalvo et al. [156] reported the inferiority of Young’s modulus and tensile strength of PLA/polyacrylonitrile (PAN) core–shell membranes, in comparison to pure PLA fibers. They attributed this poor mechanical performance to the less contact points between core (PLA) and shell (PAN) region, and weak fibers adhesion in the core–shell membranes [156]. This was caused by faster solidification of PAN as compared to PLA, due to higher polymer concentration of PAN solution. This leads to lesser contact points between PLA and PAN, and the decreased of cohesive force among the fibers, which resulted in fiber-fiber interface failures and thus, lower mechanical performance [156]. In another study, Alharbi et al. [25] highlighted the importance of chemical interactions between core and shell layer, where PLA/PVA core–shell nanofibers were observed to possess lower glass transition temperature (*T*_g_) than pure PVA and PLA fibers. They reported a single *T*_g_ value for PLA/PVA and PVA/PLA core–shell fiber systems (each at 58 and 53 °C, respectively), which indicates the strong chemical interactions between PVA and PLA chains [25]. As such, the interactions might occur due to formation of hydrogen bonding between the oxygen atom in ester groups of PLA and the hydroxyl groups of PVA [159]. This may result in changes of thermophysical properties of the core–shell systems (in this case, lower *T*_g_ value), corresponding to reduced thermal stability of the core–shell fibers and lead to softer and higher ductility fibers [25].

The enhancement of mechanical properties of core–shell fibers also was reported to be assisted by the improvement of molecular alignment in polymer chains. Jalaja et al. [49] revealed that gelatin/chitosan core–shell nanofibers possess higher Young’s modulus and tensile strength than pure gelatin fibers. They speculated that the improved tensile moduli were owing to the core–shell structure which aids the alignment of molecular gelatin chains during electrospinning. This similar underlying mechanism also was reported earlier by Merkle’s group [160]. In their study, Merkle et al. [160] reported higher elastic modulus of PVA/gelatin core–shell nanofibers compared to pure PVA fibers. It is speculated that during coaxial electrospinning, the shell solution (gelatin) protects the core region (PVA) from surface turbulence, caused by Rayleigh instability [161,162]. This protection allows the core PVA molecules to be better aligned and stretched further [160]. This eventually led to the partial transformation of PVA core into more elastic and crystallized PVA, and thus improvement of the mechanical performance. Nevertheless, despite various explanations offered in the previous studies, the exact underlying mechanism which is responsible for the mechanical properties enhancement of core–shell fibers is still highly debatable. It is anticipated that more and extensive studies at microscale and macroscale level are needed to be performed in order to reach a concrete conclusion.

Apart from physical and mechanical modifications, core–shell fibers also provide an opportunity to improve signaling or cues of the fibrous scaffolds for certain tissue engineering applications. Current myocardial treatments face challenges to provide appropriate topical cues which facilitate the formation of highly aligned myofibers. In addition, cardiomyocytes also rely heavily on electrical signaling for tissue homeostasis and consistent beating rhythms of the scaffolds [32,33]. One of the strategies to overcome these challenges is by incorporating conductive materials to modulate electrical conductivity of the scaffolds and eventually improves cardiomyocytes signal for subsequent tissue homeostasis. Liu and co-workers incorporated CNTs in PELA copolymers to achieve this goal [33]. Their work has shown that the increase in conductivity helped to maintain cell viabilities, improve production of sarcomeric α-actinin and troponin-1, and stimulate the synchronous beating of cardiomyocytes [33].

### 4.3. Preserving Sensitive Bioactive Molecules and Sustaining Their Release

Sensitive bioactive molecules are referred to molecules which have very short half-life in vivo, high volatility, or easily denatures when in contact with organic solvents. The incorporation of these sensitive biomolecules in monolithic fibers usually resulted in rapid loss of their bioactivity, hence restricting the desired therapeutic effects. The unique feature of core–shell fibers is the key solution to this limitation. The most common approach is by loading the bioactive molecules in core layer and using the shell layer as a “shield” to preserve and protect the sensitive loads from direct contact with harsh solvents during fabrication process or from rapidly-changing microenvironment during in vitro cell culture or in vivo implantation. Table 3 lists several biomolecules which are reported to be sensitive and respective core–shell fibers systems employed to preserve them.

In the case of monolithic fibers, a major concern related to possible loss of bioactivity originates from the use and interaction with organic solvents during fibers preparation, especially those prepared from hydrophobic polymers. Core–shell fibers system eliminates this apprehension by preventing direct contact between bioactive molecules and organic solvents. In addition, core–shell system also provides protection for the bioactive molecules which are sensitive to rapid changes of in vivo microenvironments. The sensitive bioactive molecules can be loaded in two ways: (1) loaded independently in the core layer without core polymer, or (2) blended or mixed with hydrophilic core (usually dissolves in common and non-hazardous solvents such as water, ethanol or bovine serum albumin (BSA)). Hydrophobic polymers can still be incorporated as shell layer in the fiber system, hence providing ideal fiber design with desired structural integrity and mechanical properties while at the same time preserving bioactivity of the loaded biomolecules and potentially controlling their release.

Although the main function of core–shell fibers is to preserve sensitive biomolecules from harsh organic solvents and rapidly changing in vivo microenvironments, the core–shell system also could work the other way around; by protecting targeted microenvironment from therapeutic yet toxic bioactive molecules. For example, paclitaxel is an antiproliferative and anticancer drug which is commonly used for drug eluting stent and its usage has been approved by the Food and Drug Administration (FDA) [166,167]. Nevertheless, the clinical application of paclitaxel is restricted due to concern over its toxicity and potentially causing serious side effects. This concern aroused from the presence of Cremophor EL in paclitaxel formulations; Cremophor EL was reported to be toxic and caused adverse effect including nephrotoxicity, neurotoxicity, and hypersensitivity [132,166,167]. Tang et al. [132] aim to improve the therapeutic effect of paclitaxel while minimizing its side effects by incorporating paclitaxel in core–shell system. In their work, ethanol-dissolved paclitaxel micromolecules noncovalently interacted with macromolecular chitosan through self-assembly, where the formation of core paclitaxel was initiated by hydrophilic/hydrophobic interaction and π–π stacking, followed by chitosan wrapping through Van der Waals forces and hydrogen bonding. In vitro studies revealed that paclitaxel/chitosan core–shell nanofibers demonstrate low cytotoxicity towards HUVECs and mouse embryonic fibroblast NIH/3T3 cells [132]. Another anti-inflammatory drug, nimesulide also has been reported to show toxicity in its hepatic metabolism [168], despite its approval to be used in treatment of serious pain states. In order to minimize its toxicity, nimesulide was incorporated in PMMA/PCL core–shell fibers system [168]. In this way, the passage through hepatic circulation can be avoided, and as a result, nimesulide can be delivered to the targeted site with minimum toxicity.

## 5. Strategies for Controllable Release of Encapsulated Bioactive Molecules

The most sought-after feature of core–shell fibers is their ability to control the release of encapsulated bioactive molecules in a sustained manner, thus restraining the undesired burst release, which typically associated with monolithic fibers. This attribute is highly desirable particularly for applications that require sustained release of biomolecules up to 30 days or longer. As briefly described in Section 2, the presence and controllable release of bioactive molecules brings huge impact to the cell growth and ultimately the desired therapeutic effects. In tissue engineering and drug delivery, the goal of controllable release is to (1) maintain biomolecules concentration at relevant dose for extended period of time, (2) eliminate concern of overexposure of biomolecules to cells and native tissue, and (3) accelerate local tissue regeneration or healing process. The first goal is particularly imperative, as therapeutic effect might not be realized if drug concentration is too low, whereas too high concentration might result in potential adverse effects. As a result, controllable release studies are extremely active research area in recent years.

Burst release is commonly observed in release profile of single or monolithic fibers due to the following reasons; (1) rapid diffusion of biomolecules from the matrix to the surroundings because of large specific area of the fibers, (2) low molecular weight biomolecules possess higher tendency for burst release due to greater osmotic pressures [169], and (3) fast degradation of polymer matrices especially those fabricated from hydrophilic polymer. Although burst release might be favorable in certain situations, for example rapid treatment, pulsatile release, or targeted delivery, it is generally undesirable due to many reasons including uncontrollable release period, poorly defined mechanism, shorter release profiles and difficulty in controlling the released dose [170]. A practical, yet simple, solution to this impasse is by introducing another layer (sheath) which completely wraps the monolithic fibers. This represents the structure of core–shell fibers. The introduction of a shell layer helps to slow down the diffusion of biomolecules (owing to longer passage distance) and decreasing the degradation rate of the polymer matrix [40,66,88].

Improved controllability of biomolecules release is made possible due to better understanding of the release mechanism. In general, the release mechanism typically involves (1) diffusion and/or (2) degradation/erosion [39,40,110]. In the cases where both diffusion and degradation/erosion are involved (e.g., biodegradable fibers), the release rate is determined by the more dominant mechanism. Diffusion typically dominates in the early stage of the drug release, while degradation/erosion is the more prominent mechanism in the later stage of the release.

The drug release through diffusion can occur in three different ways; (1) water was occupied in pores network of the fiber matrix, which by time, the pores become large enough to facilitate the release of the drug/biomolecule [170,171], (2) drug/biomolecule particles simply permeate out from the fiber matrix to the less concentrated area (i.e., the release medium or fiber surroundings) [172,173], and (3) osmotic pressure causes water influx into the fiber matrix which forces the drug/biomolecule to diffuse out from the membrane [174]. The drug diffusion from monolithic or core–shell fibers can be represented by Fick’s second law of diffusion [175]. According to this law, the drug diffusion rate is dependent on drug concentration, fiber thickness, and drug diffusion coefficient. Fick’s second law has been widely used as a basis model to predict the drug release profiles. Nevertheless, this model approach is accurate only if several conditions are met: (1) physical dimension of fiber matrix does not change during drug release process, (2) drug diffusion coefficient remains constant, and (3) drugs are distributed uniformly within the fiber matrix. These are not always the case during actual situations, and therefore other mathematical models also have been developed throughout the years to overcome these constraints, including zero-order and first-order kinetics [117,176] Hixson-Crowell [177], Higuchi [178], Korsmeyer-Peppas [179], Brazel-Peppas [180,181], Baker-Lonsdale [182], Weibull [183], Hopfenberg [184], and Peppas-Sahlin [185]. Comprehensive discussion on these mathematical models can be found in the literature [186].

Meanwhile, degradation can be defined as the scission of polymer chains which leads to formation of oligomers and eventually, monomers [187]. Erosion, on the other hand describes the material loss from the polymer as a result of monomers/oligomers leaving the polymer [187]. Based on this definition, degradation is typically monitored by the molecular weight change while erosion is usually represented by the change of weight or mass. All biodegradable polymers will eventually degrade/erode due to the presence of hydrolysable bonds in the polymer chains [188]. The only difference is how fast a polymer degrades/erodes compared to another polymer, and this is mainly influenced by the bond type in the polymer backbone which will determine the rate of hydrolysis, the main reaction of degradation [188]. The degradation trait is one of the desired criteria of tissue scaffolds as the degradable scaffold will eliminate the needs for the second surgery to remove the implants after cells growth. More importantly, degradation and erosion also play a crucial role in controlling the release rate of the drugs or biomolecules. 

Erosion can occur in two ways: (1) surface erosion and (2) bulk erosion [187,188]. During surface erosion, the polymer loses material only at the surface. In the case of core–shell fibers, although the geometric shape of fiber is maintained, the shell layer became thinner and additional macropores are introduced as a result of the surface erosion. This may increase drug release rate due to the now-shorter passage distance and the presence of macropores. The positive side is that the rate of surface erosion is proportional to surface area, thus making it highly predictable and easier to be controlled. Therefore, relevant controllable release strategies can be designed to tailor the drug release rate accordingly. Bulk erosion, in the meantime occurs when water penetrates the polymer bulk, leading to homogeneous degradation in the entire polymer. Although the molecular weight and the mechanical properties of the fiber matrix are decreased over time, the mass loss is delayed until a certain critical time point where the bulk polymer is rapidly hydrolyzed and disintegrated. Nonetheless, bulk erosion is less predictable compared to surface erosion, making it a less desired mechanism for the controllable release strategy.

Due to a better understanding of the release mechanism, the rates of biomolecules release from core–shell fibers can then be tailored by employing several strategies to achieve desirable release profile. Strategies to control the release of bioactive molecules from core–shell fibers can be categorized into two categories: (1) controlled release and (2) on-demand or smart release. Detailed discussion on both release categories is provided in the subsequent sections.

### 5.1. Controlled Release

In this review, the term “controlled release” was used to indicate the controllability of the drug/biomolecules release from core–shell fibers. The release was initiated straight-away once the fiber mats were immersed in simulated fluid (in vitro) or implanted in targeted body (in vivo). The drug release rate and mechanism are influenced by a number of different factors including fiber composition [56] and architecture [189], pore size [190], drug properties [45], and, to a certain extent, the loaded location of the drug [51]. Although some review reports [191,192] did not classify release rate control through the modification of fiber thickness or variation of composition as “controllable”; however, we are of the opinion that these modification strategies still reflect the change of controllability of release rate, and thus, classified them as controllable release strategies in this review.

#### 5.1.1. Shell Layer Thickness

The most conventional and widely-studied controlled release strategy of core–shell fibers is through tuning of shell layer thickness [52,110,111,189]. In thicker shell layer, the core-incorporated biomolecules have to diffuse through longer distance, resulting in slower release. Faster release can be achieved, vice versa, through a thinner shell layer. Thicker shell layer can be obtained via several approaches including increasing shell flow rate [67,189], reducing core flow rate [52] and using higher concentration of shell solution [87,110,193]. To give an example, Liu et al. [111] prepared ferulic acid-loaded gliadin/CA core–shell nanofibers by triaxial electrospinning, where solvent mixture of acetic acid and acetone was used as outer working solution. Various core–shell nanofibers with different shell thickness were prepared by varying intermediate (CA solution) flow rate during electrospinning. Three different intermediate flow rates were employed: 0.1 mL/h (F2), 0.2 mL/h (F3), and 0.5 mL/h (F4), resulting in shell thicknesses of 5.2 ± 2.6, 14.7 ± 1.6, and 30.2 ± 10.1 nm, respectively (TEM images were shown in Figure 9a). A gliadin monolithic fiber (F1) was also prepared as control.

As depicted from the release profiles of ferulic acid in Figure 9b, the release rate was reduced gradually with the increase of shell thickness [111]. In the first 15 min, the cumulative release of ferulic acid for F1, F2, F3, and F4 was observed to be 20.3% ± 5.1%, 1.5% ± 1.2%, 0%, and 0%, respectively, whereas the cumulative release is 37.2% ± 4.5%, 15.4% ± 4.6%, 7.6% ± 4.1%, and 3.1% ± 2.2%, respectively, after one hour release. All core-shell samples (F2, F3, and F4) demonstrate sustained release with no obvious initial burst release. In contrast, significant initial burst release was clearly observed from gliadin monolithic fibers [111]. 

Meanwhile, contradictory observations also have been reported by other research groups. They still observe the initial burst release in two-stage release from the core–shell fibers. To give an example, icariin was shown to be successfully released in controlled manner from icariin/SF-poly(lactide–*co*–ɛ-caprolactone) (PLCL) core-shell fibers, in which the entire release period included two phases: an initial burst release phase (∼47.54% ± 0.06% of icariin was released in first 5 days), followed by consistent release phase (icariin release accumulated to 82.09% ± 1.86%) afterwards [39]. The consistent release mechanism was associated to diffusion through adjustment of shell layer thickness, and polymer degradation which lead to the presence of pores, and as a result triggering the scouring effect of dissolution medium on fiber surface. 

The sustained release of icariin ensures the drug was delivered at an effective concentration, which led to the positive osteogenic differentiation of bone marrow-derived mesenchymal stem cells (BMMSCs). This was evidenced by the obvious mineralized deposits via Alizarin Red staining (Figure 10a), and the increased ALP activity (Figure 10b) after 14 days of cultivation [39]. Further in vivo studies revealed that faster bone formation which covers most of bone defect region (Figure 10c) as well as higher bone volume (Figure 10d) and density (Figure 10e) was observed in rat calvarial defects after implanted with icariin-containing fiber mats [39]. Tailoring shell layer thickness had been one of the promising strategies to control biomolecules release. However, despite extensive progress and development, the thickness consistency and difficulty of ultra-fine tuning of shell thickness remains as major challenges in this release strategy.

#### 5.1.2. Shell Layer Composition

Another extensively studied strategy for controllable release from core–shell fibers is through variations of shell layer composition. Varying shell composition affecting biomolecules release profiles through many ways including changes of degradation rate, diffusion rate, as well as water absorption capacity. In the first case, the release profile was tuned by adjusting shell degradation rate through material selection or material blends. For example, Liu et al. [59] reported two different release profiles of GDNF and NGF by incorporating them in PLGA- and PDLLA-based core–shell fibers, respectively. Both fiber scaffolds showed initial burst release within 1 day, followed by sustained release afterwards; 20.3% GDNF was released cumulatively in the first 24 h, and the release increased gradually to reach 62.5% after 42 days. Meanwhile, slower release of NGF was observed: cumulative release was recorded to be 12.4% within 24 h and only 26.5% after 42 days. The slower release of NGF was associated with higher hydrophobicity and lower degradation rate of PDLLA compared to PLGA [59].

Shell degradation rate is also typically adjusted through blending of two or more polymers; in most cases, hydrophobic polymer was blended with hydrophilic polymer where the content of one of them will be varied to obtain the desired degradation rate. For instance, PLCL was blended with collagen and chitosan to form the shell layer of core–shell fibers. It was demonstrated that for fiber with higher content of PLCL, the release rate of incorporated heparin was lower due to slower degradation of the shell layer [56]. In another study, different content of poly(ethylene glycol) (PEG) was added to PLGA (5%, 10%, and 15% of PEG respective to PLGA weight), where this PEG-PLGA blend was used as the shell material. It was found that higher PEG content led to faster shell polymer erosion and hence, faster release of BSA from the core compartment [194]. Similar observation (i.e. faster biomolecule release) was reported in other studies for fibers with higher collagen [61] and SF [103] content.

In the second case, the modification of shell layer composition affecting release profiles through the change of diffusion rate. For instance, different PEG concentrations (0.25%, 1%, and 3%) were added to PCL shell in recombinant human vascular endothelial growth factor (rhVEGF)-incorporated PEO/PCL-PEG core–shell fibrous mats [190]. The research group reported that the addition of 3% PEG resulted in mean pore size doubled to that observed in 1% PEG fiber. Greater pore size and higher pore density facilitates faster rhVEGF release since the release relies on effective diffusion. However, surprisingly lower total release of rhVEGF was observed from 3% PEG samples compared to 1% and 0.25% PEG scaffolds. This observation was attributed to the adsorption of rhVEGF to PCL which may occur in high PEG-containing systems [190]. The results indicate that drug release profiles may depend on more than one factor, and future release strategies might have to take into account the factor of drug affinity towards different polymer or system.

In the third case, the release profiles were affected by the changes of water absorption capacity due to alteration of shell layer composition. In this study, PCL-nHAP was used as core layer while PCL or PVAc were examined as different shell materials [107]. BMP-2 was encapsulated in PLGA nanoparticles which were electrosprayed onto the surface of the core–shell fibers. For both shell materials, burst release of BMP-2 was observed in the first 8 h, followed by sustained release in the course of 28 days. However, PVAc shell demonstrated lower burst release (40% in 8 h) than PCL shell (68% in the same period). Better restraining of burst release was associated to high absorption capacity of PVAc (350% water absorbed in 1 day) which might help to retain more BMP-2 that been released from PLGA nanoparticles during swelling [107].

#### 5.1.3. Drug Concentration, Properties and Loaded Location

Apart from fiber properties and architecture, drug release profiles from core–shell fibers are also very much dependent on drug concentration, drug properties (e.g., molecular weight and affinity towards polymer or system), and drug-loaded location. For the first case, majority of studies reported faster drug release with higher drug concentration [36,52,79,113,119,128,195]. The reason was related to the release behavior which was governed by diffusion law. As a consequence, higher drug concentration leads to more intense diffusion, which resulted in higher release rate. However, contradictory observation also has been reported. Yin et al. [56] reported slower heparin release for higher drug concentration in 45 days release study; 5%, 15%, and 30% loading resulted in real heparin loading amount of 6.8, 7.9, and 10.5 mg, respectively. Over 45 days, the 5%, 15%, and 30% heparin loading leads to cumulative release of 96%, 80%, and 61%, respectively [56]. The heparin was loaded in the core, enclosed by the shell layer comprised of collagen, chitosan and PLCL. Although it was not clearly stated, the contradictory observation was highly probably caused by easier fluid penetration into core compartment with lower heparin molecules available, and thus higher drug release rate. 

For the second case, drug release rate was reported to be influenced by drug properties. Mistry et al. [45] incorporated two protein models of different molecular weight (human insulin (6.6 kDa) and human immunoglobulin G (IgG) (150 kDa)) in collagen-Matrigel-GelMA/PEGDA-alginate core–shell hydrogel strands. It was revealed that IgG was more likely to remain in the strands due to its higher molecular size [45]. In another study, two bioactive compounds; heparin and salvianolic acid B were loaded in core, and shell layer, respectively [58]. For the case of salvianolic acid B, the natural extract was incorporated in mesoporous silica nanoparticles (MSNs) prior mixing with other shell materials (i.e., PLCL and collagen type I). In vitro release study revealed that heparin was released faster than salvianolic acid B despite the fact that heparin was loaded in core layer. This observation was owing to the greater effect of salvianolic acid B affinity towards MSNs compared to the effect of diffusion-induced heparin release from the core [58].

Meanwhile, for the third case, the release rate was affected by the loaded location of the drug. Generally, this involves dual delivery or even multiple drug delivery. For dual delivery, two different biomolecules are loaded in core and shell layer separately [51,71,195,196,197,198]. Bioactive molecules in the core layer generally are released slower than the one loaded in shell layer: the molecule in core layer has to diffuse in longer passage distance than the molecule in shell layer. Dual delivery is useful for application that requires two distinct biomolecules release profiles. For example, magnesium l-ascorbic acid 2 phosphate (MAAP) and salvianolic acid B were incorporated in shell and core layer, respectively to achieve sequential release of the biomolecules for cardiac tissue regeneration [198]. MAAP was designed to be released first due to its ability to stimulate proliferation of cardiac cells, hence the reason of why it was loaded in the shell layer. While core-loaded salvianolic acid B was intended to be released later owing to its capability to promote and induce cardiac cells differentiation [198]. 

In another work, bacterial inhibitor-metronidazole and tissue regeneration enricher-naringin were incorporated in shell and core compartment of PVP/PLGA core–shell fibers, respectively [51]. Initial release of metronidazole effectively inhibits anaerobic bacteria colonization in 3 weeks while later release of naringin promotes proliferation of mouse osteoblasts cells (MC3T3-E1), hence rendered the core–shell fibers as an effective scaffold for guided tissue regeneration with infection control [51].

### 5.2. On-Demand Release

Advanced biomolecules release system triggered by internal or external stimuli is attracting ever increasing attention from the research community [199,200,201]. This release system is widely referred as on-demand release or smart release [82,202,203,204]. The drug/biomolecules release can be initiated or stopped in response to the triggering of a stimulus. Although the scale of on-demand release studies from core–shell fibers is yet to match the scale of core–shell particles or spheres, the extensive studies of smart release involving core–shell fibers is expected to accelerate considering their potential as drug delivery tools for therapeutic tissue regeneration. Several stimuli-triggered release systems involving core–shell fibers are discussed in this section including pH-, temperature-, ultrasound-, and light-stimulated release system.

#### 5.2.1. pH-Stimulated Release

A pH-responsive polymeric material is capable of transferring protons in an event of environmental change of pH. A change of pH induces electrostatic repulsion in aqueous solution triggered by ionic interactions, and this leads to the collapse of the polymer chain. The most common and widely-investigated pH-responsive material is Eudragit, a trade name for copolymers that consist of different ratio of methacrylic acids and acrylic esters. Yang et al. [108] incorporated diclofenac sodium, a nonsteroidal anti-inflammatory drug in lecithin/Eudragit S100 core–shell nanofibrous matrices. The drug was virtually unreleased at pH 1.0, where cumulative release was only 2.8% after 2 h, in contrast to 79.1% release in dissolution media of pH 7.0 after 22 h. This release profile was due to the employment of Eudragit S100 as shell layer, where this polymer is insoluble at acidic condition [108].

A similar observation is reported in another work, where two different drugs (i.e., indomethacin (IMC) and mebeverine hydrochloride (MB-HCl)) were loaded in PEO/Eudragit S100 core–shell fiber system [205]. In vitro drug release study revealed that for both drug cases, only minimal release was observed at pH 1.2, while rapid release was obtained in buffer of pH 7.4. This showed that Eudragit S100 shell layer can effectively stop drug release at pH below 7. After 6 h release at pH 7.4, between 65.1 and 78.7% release of IMC was recorded while MB-HCl was released in the range of 62 to 94.6% in the same period. The authors also reported that higher drug content resulted in lower drug released after 8 h. This was associated to the decrease in polymer presence in neutral condition to aid drug solubilization when the drug content increases [205]. Although the drug release was initiated by the change of pH, other studies have discovered that the release rate is still governed by the shell layer thickness [43,206] and shell polymer concentrations [207].

Apart from polymer dissolution, change of pH may also affect release profiles through alteration of drug solubility [208] and distribution [209]. For instance, curcumin, a hydrophobic drug, was loaded in hydroxypropyl-beta-cyclodextrin (HPbCD)/PLA core–shell nanofibers, and its release were examined in two pH conditions: pH 1 to represent simulated gastric fluid, and pH 7.4 to epitomize simulated intestinal fluid. Curcumin can exist in various forms; cationic, anionic and neutral forms where these forms are pH-dependent. In acidic condition, curcumin exists in cationic form and this led to the increased solubility of curcumin, and thus faster release compared to neutral condition [208].

Meanwhile, pH change also was reported to influence drug distribution in fibers. Bevacizumab in two different pH buffers (i.e., pH 6.2 and pH 8.3) were enclosed by PCL shell using coaxial electrospinning. In vitro release study revealed that core–shell fibers loaded with bevacizumab at pH 6.2 exhibited faster release (cumulative release of 60.6% ± 7.3% over 19 days) than those loaded at pH 8.3 (55.6% ± 16.8% release over 60 days) [209]. This observation can be explained by the following mechanism; bevacizumab has a net positive charge at pH 6.2, which forces it to migrate from the core to the shell layer. Upon exposure to shell solvent, bevacizumab becomes unfolded and mixed with intact bevacizumab in the core layer which ultimately resulted in first-order release kinetic (faster release) (Figure 11a). In contrast, bevacizumab stays neutral at pH 8.3 (isoelectric point of bevacizumab) and as a result, it remains in core layer with sustained release profile (Figure 11b) [209].

#### 5.2.2. Temperature-stimulated Release

Temperature-stimulated release is another release strategy currently attracting increasing attention for smart release system in drug delivery and tissue regeneration. In general, temperature-responsive material allowing polymer chain extension or contraction in response to small changes of temperature. Poly(*N*-isopropylacrylamide) (PNIPAAM) is the most extensively used temperature-sensitive material due to its unique feature; PNIPAAM can undergo phase change from hydrophilic at 25 °C (below lower critical solution temperature (LCST)) to hydrophobic structure at temperature above LCST in a reversible manner. To give an example, the release of a hydrophobic drug model, combretastatin A4 from PLA/PNIPAAM core–shell fibers was investigated at two different temperatures: 25 and 40 °C [48]. At 25 °C (below LCST), PNIPAAM chains in hydrophilic structure were extended and as a result, release medium can penetrate into the fiber interior and release some drugs. However, 50%–60% of the drug remained unreleased due to strong affinity between hydrophobic combretastatin A4 and PLA core. At 40 °C (above LCST), hydrophobic PNIPAAM chains were contracted, resulted in shell deformation which led to cumulative release of ∼70% after 10 h [48]. 

The LCST of PNIPAAM is able to be modified though by adding co-monomers or other units to its polymer chain. For instance, Wei et al. [82] increased the LCST of PNIPAAM from 32 to 43 °C by adding acrylamide (Am) monomer via radical copolymerization. The release of curcumin from polyethersulfone (PES)/poly(*N*-isopropylacrylamide–*co*–acrylamide) (PNIPAAM–*co*–Am) was examined at 20 and 60 °C for potential application as drug delivery for cancer thermotherapy. In vitro release study demonstrated that the curcumin release is slower at 20 °C (cumulative release of 33.21%), compared to release at 60 °C (cumulative release of 80.15%). The higher release at 60 °C was due to shell deformation which opens some pores and hence, allowing curcumin release at faster rate [82].

#### 5.2.3. Other On-demand Release Strategies

Apart from widely investigated pH- and temperature-stimulated release, other on-demand release strategies also have been reported in the recent years. One of interesting example is drug release strategy through ultrasound sonication. Silica nanoparticles were attached onto the surface of rhodamine B-incorporated PEO/PLA core–shell nanofibers through parallel coaxial electrospinning plus electrospraying (Figure 12a) (detailed fabrication method was described in Section 3.1). The nanoparticles were then embedded into the shell layer via solvent–vapor annealing at 38 °C [106]. The embedding depth was shown to be a function of annealing time where 30 minutes annealing embedded roughly half of the nanoparticles (Figure 12b). The subsequent ultrasonication causes silica nanoparticles to detach (uncork) and create openings which triggered rhodamine B release from PEO core (Figure 12c). The release profiles of rhodamine B is shown in Figure 12d. For samples without annealing, high release rate was observed (2%/hour release over 10 h). Upon 30 min sonication for 24 and 36 h, the release rate was further increased to 6%/hour and 7%/hour, respectively. Although annealing appears to successfully slow down the drug release, the release rate can still reach to 5%/hour release [106]. To fine-tune the release of the drug, extra works may need to be performed with particular focus on wider range of annealing and ultrasonication time.

Another fascinating release strategy which was recently published is near-infrared (NIR) light-triggered release. Choi et al. [118] encapsulated doxorubicin (DOX) hydrochloride, an anticancer drug and fluorescein isothiocyanate-labeled bovine serum albumin (FITC-BSA), in the core of PLGA hollow fibers. Ligand exchanged gold nanorods (AuNRs), a photothermal agent were incorporated in the shell layer to create NIR light-triggered release system. Upon exposure to NIR light, AuNRs generated heat, which increases fiber local temperature [118]. When the temperature raised above the glass transition temperature (*T*_g_) of PLGA, the polymer chains became mobile and free volume within shell was enlarged. As a consequence, drug was rapidly release from the fibers. Vice versa, when the NIR light was turned off, local temperature was dropped, and segmental motion of polymer chains was stopped, resulting in terminated drug release (Figure 13a). The local temperature can be varied easily by altering power density of NIR light (Figure 13b). 0.4 W/cm^2^ NIR light power density allowed approximately 3.5% of DOX release at each light exposure (Figure 13c). During the release test, no morphological and mass changes were observed indicating the drug release was not occurred due to the PLGA degradation [118]. On-off action of NIR light can trigger the segmental switchability of polymer chains, and this led to accurate and prominent on-demand drug release.

## 6. Conclusions and Future Perspectives

Although monolithic fibers are able to imitate architecture and mechanical properties of native ECM, further clinical advancements have been restricted by their inability to protect the bioactivity of incorporated sensitive molecule and sustaining its release. The presence of bioactive molecules has been shown to bring positive impact to survivability and metabolic activity of cells, provided their bioactivity is preserved and their concentration is maintained at therapeutic level. Only then, the successful clinical applications could be realized. Core–shell fibers emerge as an outstanding upgrade to monolithic fibers, in which they solved almost every limitation associated with their monolithic counterparts. The development of core–shell fibers has progressed at an accelerated pace, where a very broad selection of material combinations for core and shell layer has been reported particularly in the last five years. The core–shell material combinations include synthetic/synthetic polymer, synthetic/natural polymer, natural/synthetic polymer, and carbon-based material/synthetic polymer combination. The material selection, each for the core and shell layer is the most critical step in designing a core–shell fiber. It can be concluded that the final materials selection usually depend on these factors: (1) desired physicochemical, mechanical and biological properties of the core–shell fiber, (2) the nature of drug/biomolecules to be incorporated, and (3) the corresponding release strategies of the incorporated biomolecules.

We have summarized that two design configurations, which typically adopted during preparation of core–shell fibers (Figure 1a): (1) hydrophilic polymer and/or bioactive molecule employed in the core region, with hydrophobic polymer selected as shell layer, and (2) hydrophobic polymer and/or bioactive molecule incorporated in the core layer, with hydrophilic polymer chosen as the shell material. Yet, each of the design approach has its own constraints. Although the first design configuration shows superior controllability of biomolecules release (which is perfect for drug delivery applications), the hydrophobic shell has low affinity towards cells owing to its hydrophobicity and low cell recognition sites, which may hinder its applicability in tissue regeneration. As a result, the core–shell fibers adopting the first design approach usually accompanied by post-fabrication or surface treatments. Utilizing blends of hydrophobic/hydrophilic polymer is another common approach to overcome the low cells adhesion issue of these fibers. Meanwhile, the core–shell fibers adopting the second design configuration suffer from burst and faster release of biomolecules, due to rapid degradation and erosion of the hydrophilic shell layer. Therefore, the core–shell fibers prepared through this design approach usually cross-linked to enhance the structural and mechanical properties of the fibers, as well as to slow down the degradation and erosion process.

In term of fabrication strategy, coaxial electrospinning currently is the most extensively studied method for the preparation of core–shell fibers primarily due to its relatively simple setup, flexibility for tuning of process parameters, and ability to process wide-range of materials. Further advancement of coaxial electrospinning has quickly followed where vast reports on triaxial electrospinning, needleless coaxial electrospinning, coaxial electrospinning plus electrospraying, coaxial electrospinning with micropatterned collector, and magnetic field-assisted coaxial electrospinning have emerged in the recent years. Despite its outstanding progression, coaxial electrospinning is not without limitations. Major constraints currently associated with coaxial electrospinning are low throughput of fibers formation, difficulty in obtaining 3D architecture of core–shell fibers, and the use of high voltage and conductive collector which might be unsafe for the operators. These motivate other research groups to explore on the alternative methods for the fabrication of core–shell fibers which include CEHD direct printing, coaxial bioprinting, self-assembly, emulsion centrifugal spinning, solution blow spinning, coaxial airbrush, and microfluidics. Although these alternative fabrication strategies can solve some limitations associated with coaxial electrospinning, they are also not without their own restrictions. These include large size of ~1 µm fiber diameter [133], and difficulty in future development of aligned or patterned core–shell fibers. Therefore, we anticipate more works will be done in this area, largely directed towards further modifications of coaxial electrospinning and exploration of novel fabrication technique for the core–shell fibers.

Although the main roles of core–shell fibers are to provide better preservation of sensitive bioactive molecules and superior controlling of biomolecules release, these roles have further expanded which now also include allowing fiber formation of almost any materials and permitting flexible modification of physical and mechanical properties of the fibers. Core–shell structure, for instance has been shown to enable formation of fibers from materials which are possessing poor spinnability such as PGS, lecithin, and some natural polymers (e.g., collagen, SF, and zinc). This role is particularly imperative as it may open-up possibilities to fabricate novel pristine fibers from other reportedly promising yet unspinnable material such as conductive polymer, or as an alternative strategy to prepare ultrafine fibers of bioactive glass and graphene with diameter less than 100 nm (current reported diameter of bioactive glass and graphene fibers stands at 100–800 nm [210,211] and 1–150 µm [212,213], respectively). Conductive polymer, in particular is difficult to be formed in fiber structure due to inadequate chain entanglement (causes by its rigid backbone and lower molecular weight) and unstable jet formation (causes by high solution conductivity) [214,215]. Therefore, the establishment of core–shell structure may allow the formation of conductive polymer-based pristine fibers which will be of great interest for electrical signaling-dependent tissue engineering such as cardiovascular regeneration.

The advancement of core–shell fibers also enables flexible modification and improvement of physical and mechanical properties of the fibers, where two immiscible polymers or materials can now be combined together in a single core–shell fiber without compromising its physical and mechanical capabilities. This is highly attractive as unlimited range of materials can be employed to achieve the desired physical and mechanical enhancement of the fibers. Although the mechanical improvement was clearly observed in core–shell fibers compared to monolithic fibers, the exact underlying mechanism which responsible for this improvement is still inconclusive. This will pave the way for more micro- and macroscale studies to be conducted in the future to seek the concrete conclusion for this mechanism. Core–shell structure also was shown to play a crucial role in tissue engineering and drug delivery by protecting sensitive biomolecules (e.g., drug, protein, cells, and natural extract) from harsh solvents during fiber fabrication or from rapidly-changing in vivo microenvironments post implantation. Interestingly, the core–shell structure can also function in an opposite manner, by shielding the targeted microenvironments from therapeutic yet toxic biomolecules. Again, this will allow almost any therapeutic biomolecules to be incorporated in the fibrous scaffolds, which may ultimately lead to the closest resemblance of the native ECM. Sustainable and controllable release of biomolecules also was made possible through the employment of core–shell fibers. A number of controllable release strategies have been discussed comprehensively in Section 5, where we classified these strategies into two categories; (1) controlled release which is dependent on physicochemical properties of the fibers and the loaded-drug properties, and (2) on-demand or smart release which triggered in response to internal (e.g., pH) or external (e.g., temperature, ultrasonication, and NIR light) stimuli. The trend of incorporating multiple cues or biomolecules are gaining ever increasing attention with release strategies are directed towards sequential on-demand or on–off switch release profiles. It is worth mentioning that the smart release studies of core–shell fibers are still not as extensive as the works done for the core–shell particles or spheres. Therefore, the on-demand or on-off switch release profiles are still not well-defined and further investigations on various stimulated releases from core–shell fibers are anticipated to be performed to rectify this issue.

Special mention on the current trend in core–shell fibers design pointing towards the use of natural polymers as core and shell materials, primarily due to their greatest advantage of being “green”, that is synthesized from natural and renewable resources, and can be processed using environmental-friendly solvents. The natural polymers such as chitosan, collagen, and gelatin are reported to possess excellent biodegradability and biocompatibility almost similar to ECM components, which favors their applicability in tissue engineering and drug delivery. Despite these obvious advantages, natural polymers also possess inherent limitations including broad distributions of molecular weight, and significant batch-to-batch variability. In addition, most natural polymers lack mechanical strength by themselves, and typically requires cross-linking or blending with synthetic polymers to improve their respective mechanical performance. All these limitations make them less attractive compared to synthetic polymers which are more versatile and easily reproducible with tailorable properties. Nonetheless, as the current awareness is progressing towards “green” and environmental-friendly approaches especially in biomedical field, the studies on core–shell fibers utilizing natural polymers are anticipated to grow even more in the foreseeable future with more works are expected to be conducted in the area of fiber design, physical, and mechanical optimization of the fibers, and theoretical predictions of the properties of natural polymer-based core–shell fibers.

In this review, we have discussed myriad fabrication strategies, numerous core–shell material combinations, various biomolecules-containing core–shell fibers, and extensive in vitro/in vivo testing of core–shell fibers. Despite these advancements, not a single clinical trial of core–shell fibers has been reported at the time of review writing where zero result was retrieved for search with keyword “core shell fibers” from clinicaltrials.gov. This might be due to many core–shell fibers-related challenges which are still needed to be overcome. One of the challenges is larger core–shell fibers diameter compared to monolithic fibers counterpart; where even less than 1 nm diameter of monolithic fibers has been reported recently [216]. Smaller diameter of nanofibrous scaffolds is desired as it allows better cells-biomaterial interactions (due to large surface area) and good delivery of oxygen uptake and byproducts removal [217]. Coaxial electrospinning is the most likely approach to produce <50 nm diameter of core–shell fibers, which is the realistic target at present. Nevertheless, this may require extensive optimization procedures, involving optimization combinations of many process and solution parameters. In addition, the ultrafine core–shell fibers might be produced only from certain type of material, for example polymer with higher molecular weight. The reason is this polymer might possess sufficient chain entanglements at very low solution concentration, which is crucial in the preparation of fibers with ultra-fine diameter sizes.

Secondly, it is challenging to form 3D nanofibrous scaffolds with core–shell architecture. While 3D core–shell scaffolds can be fabricated through coaxial bioprinting, they often consist of larger fiber diameter and lack physical integrity and mechanical strength required for tissue engineering and drug delivery. Therefore, alternative solutions for fabrication of 3D core–shell nanofibrous scaffolds are needed urgently and this could be possibly achieved through layer-by-layer stacking method or through the employment of 3D patterned collector during coaxial electrospinning. On the other hand, future development of aligned and patterned fiber may also affect core–shell architecture, and this warrants further detailed investigations.

Thirdly, majority of release studies of core–shell fibers are evaluated in vitro. The release profiles may be affected by the complex microenvironments of in vivo, and the lack of correlation between in vitro and in vivo release profiles may hinder the applicability of core–shell fibers in ultimate clinical applications. Thus, this is another area which requires further investigations with the main goal to narrow the gap between in vitro and in vivo release profiles from core–shell fibers.

Last but not least, most of in vivo testing of core–shell fibrous scaffolds is performed using healthy animals. However, this is not the reflection of the real clinical demand. For example, patients requiring cardiovascular implants may also associate with other secondary diseases such as diabetes and hypertension [218,219], and this should be taking into account when designing in vivo animal models. Hence, more in vivo studies involving unhealthy animals are needed to be conducted in order to facilitate the potential clinical trial of core–shell nanofibrous scaffolds in the future. This is because in vivo studies are widely regarded as preclinical trial and their results are often demanded by the regulatory bodies prior clinical trial commencement. Nevertheless, it is also worth to mention that in vivo testing costs a lot [220] and huge number of animals are sacrificed for the testing, where there is no guarantee that positive in vivo results will lead to successful clinical trial afterwards. The reason is simply because each different individual may respond differently to the tissue scaffolds compared to the in-lab tested animals (e.g., mice or rabbit) due to diversified genetic, biomechanics, and immunological response of the patients. Therefore, to eliminate all these concerns we envisage that the way forward towards clinical applications might involves integration of smart core–shell fibers with the most recent advanced stem cell therapy, called induced pluripotent stem cells technology [221,222].

To conclude, recent advancements have shown that core–shell fibers is on the right path to fulfill its potential as one of the most promising structures that closely resemble native ECM. Although there are many core–shell fibers-related challenges that still need to be overcome, the glimpse of promising sign is there, and we might see the first clinical trial of core–shell fibers in the not too distant future. 

## Figures and Tables

**Figure 1 polymers-11-02008-f001:**
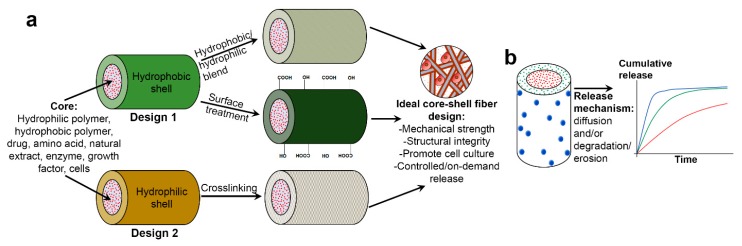
(**a**) Scheme illustrating two different design approaches of core–shell fibers and (**b**) their role in multiple biomolecules delivery with controllable release profiles.

**Figure 2 polymers-11-02008-f002:**
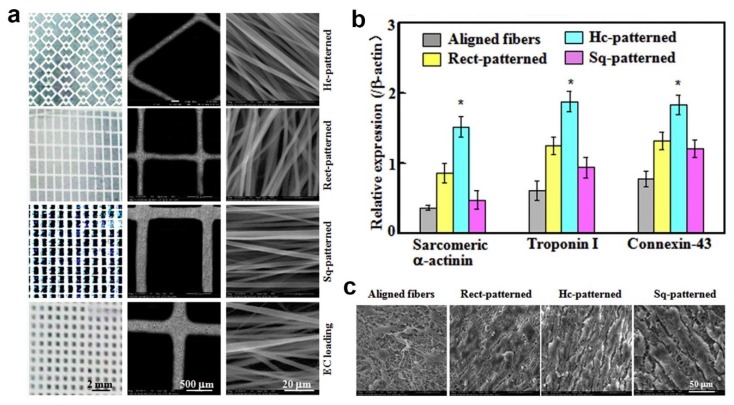
(**a**) Digital and SEM images of fibrous mats with Hc-, Rect-, and Sq-patterns for CM and EC loadings and fibers coated on patterned-mats strut. (**b**) Sarcomeric α-actinin, troponin I and connexin-43 quantification analyses from Western blot image band densities (relative protein levels were normalized against β-actin signals, n = 3; *: *p* < 0.05 compared to other groups). (**c**) SEM images of cellular morphologies post 15-day CMs coculture on aligned, Hc-, Rect-, and Sq-patterned fibrous mats. Reproduced from the work in [32]. Copyright 2017 with permission from Elsevier.

**Figure 3 polymers-11-02008-f003:**
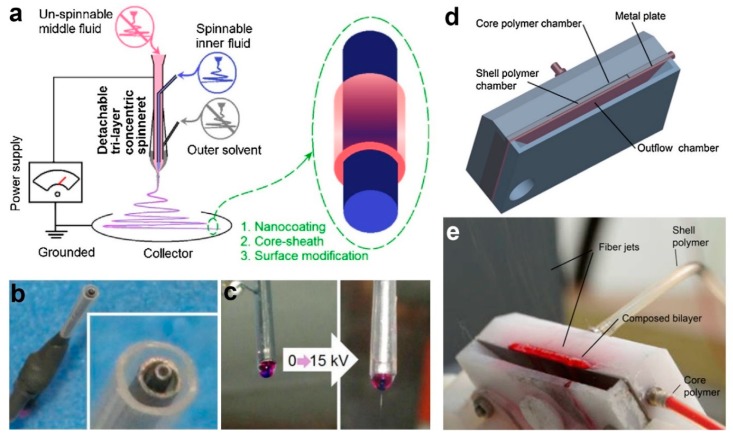
(**a**) Schematic diagram of modified triaxial electrospinning setup. Digital images of (**b**) the triaxial concentric spinneret and (**c**) the droplet shape before and after 15 kV voltage was supplied. Reprinted from the work in [110]. Copyright 2019 with permission from Elsevier. (**d**) Schematic design of needleless coaxial spinneret and (e) digital image of bilayer polymer jets on needleless spinneret. Reprinted from the work in [95]. Copyright 2017 with permission from Elsevier.

**Figure 4 polymers-11-02008-f004:**
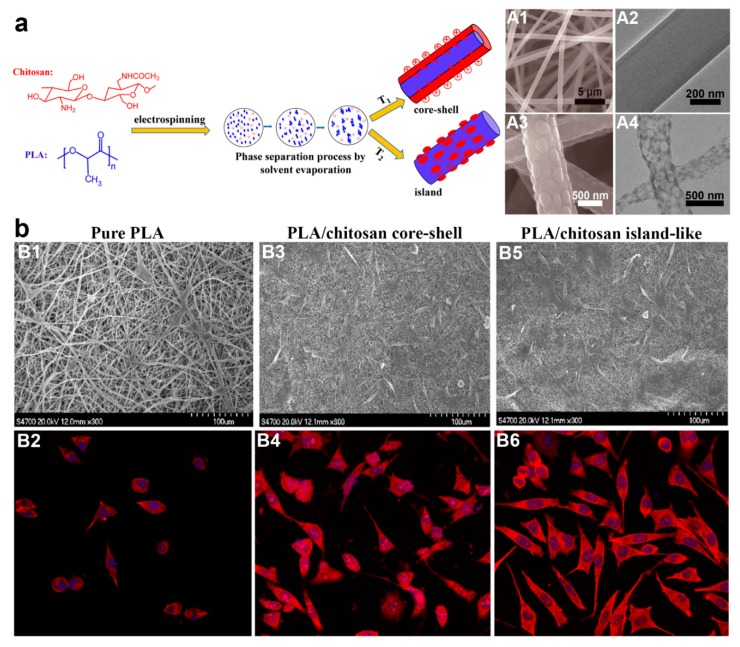
(**a**) Scheme illustrating the preparation of core–shell and island-like fibers through single electrospinning plus in situ phase separation. SEM and TEM images of core–shell and island-like structures are shown in A1, A2, and A3, A4, respectively. (**b**) SEM and laser scanning confocal microscopy (LSCM) micrographs of mouse preosteoblasts after 48 h cultivation on pure PLA fibers (B1, B2), PLA/chitosan core–shell fibers (B3, B4), and PLA/chitosan island-like fibers (B5, B6). Prior LSCM testing, mouse preosteoblasts were stained with tetramethylrhodamine isothiocyanate (TRITC)-labeled phalloidin and 4′,6-diamidino-2-phenylindole (DAPI). All LSCM micrographs are at 400× magnification. Reproduced from [125]. Copyright 2017 with permission from American Chemical Society.

**Figure 5 polymers-11-02008-f005:**
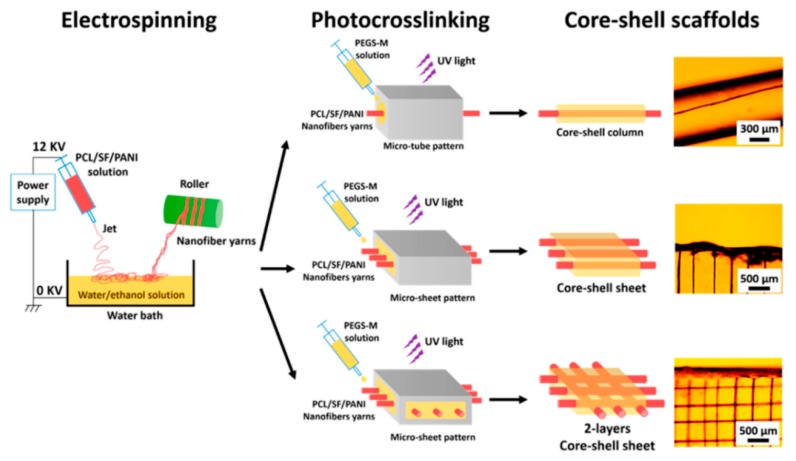
Schematic illustration of core–shell fibers preparation via single electrospinning plus UV photocross-linking. Reprinted from the work in [68]. Copyright 2015 with permission from American Chemical Society.

**Figure 6 polymers-11-02008-f006:**
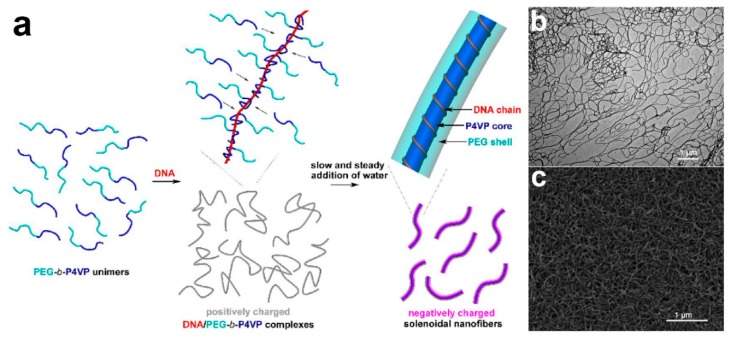
(**a**) Schematic illustration of P4VP/PEG core–shell nanofibers preparation via highly kinetics-controlled DNA/polymer self-assembly. (**b**) TEM and (**c**) FESEM micrographs of 1,4-dibromobutane-cross-linked P4VP/PEG core–shell nanofibers fabricated through self-assembly of PEG113–*b*–P4VP67 block copolymers with calf thymus DNA. Reprinted from the work in [131]. Copyright 2018 with permission from American Chemical Society.

**Figure 7 polymers-11-02008-f007:**
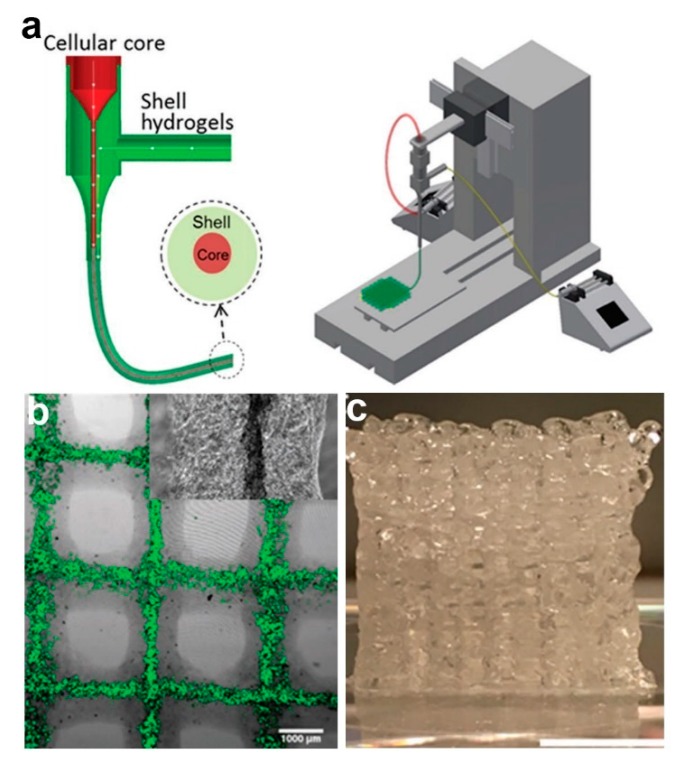
(**a**) Schematic diagram portraying the coaxial spinneret (**left**) and bioprinter setup (**right**). (**b**) Fluorescent-labeled cells in core–shell bioprinted 3D lattice (1 mm scale bar). (**c**) Digital image of 20-layer bioprinted construct (10 mm scale bar). Reprinted from the work in [45]. Copyright 2017 with permission from Wiley-VCH.

**Figure 8 polymers-11-02008-f008:**
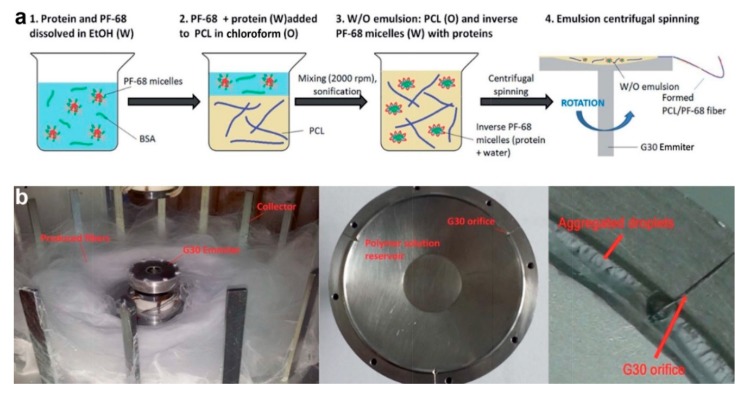
(**a**) Schematic illustration of emulsion preparation process. (**b**) Digital images of centrifugal spinning setup. Reprinted from the work in [50]. Copyright 2017 with permission from Royal Society of Chemistry.

**Figure 9 polymers-11-02008-f009:**
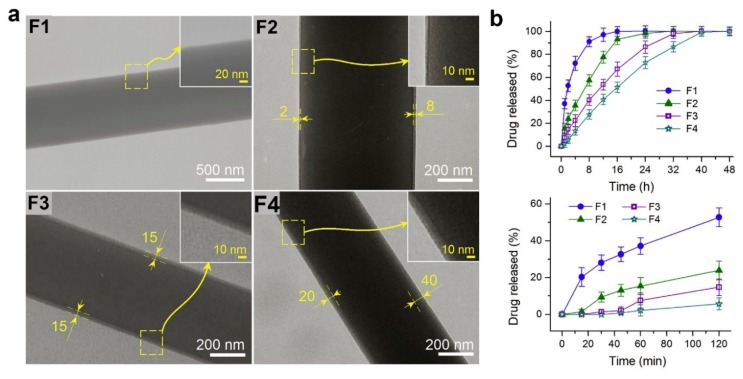
(**a**) TEM images of monolithic fiber (F1) and core–shell fibers with different shell thickness (F2, F3, and F4). (**b**) Cumulative release of ferulic acid from F1, F2, F3, and F4 in 48-h release study. Reprinted from work in [111]. Copyright 2019 with permission from Elsevier.

**Figure 10 polymers-11-02008-f010:**
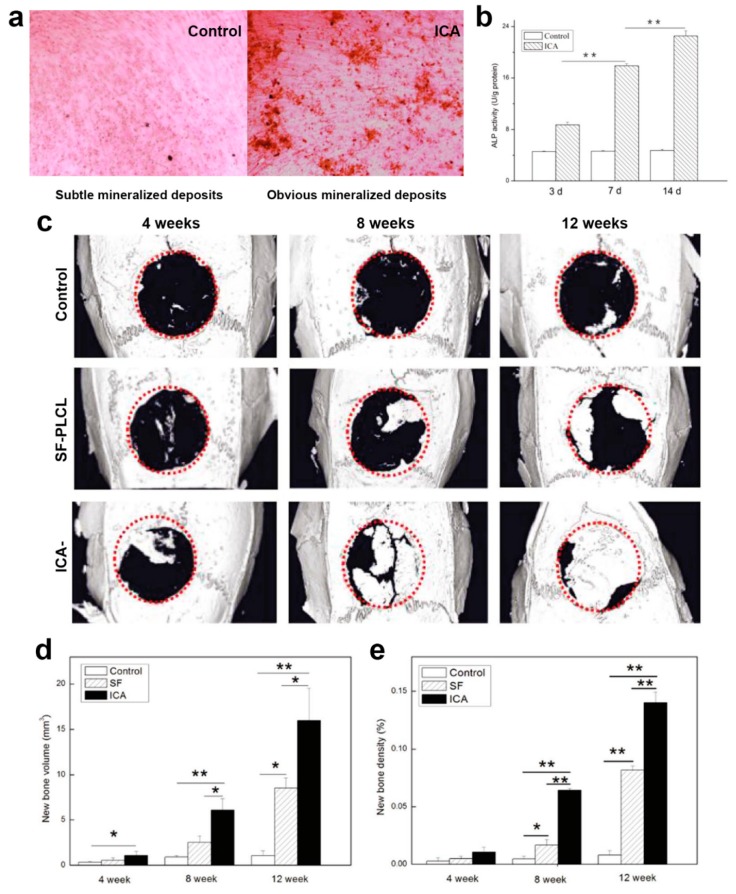
(**a**) Alizarin Red staining (after 14 days) and (**b**) ALP activity assay of bone marrow-derived mesenchymal stem cells (BMMSCs) incubated in icariin/SF-PLCL fiber mats released medium. (**c**) Micro-computed tomography (µ-CT) images of rat calvarial defects and percentage quantification of new bone volume (**d**) and density (**e**) after 3-month fiber mats implantation. ICA denotes icariin/SF-PLCL core–shell fiber mats while SF indicates SF-PLCL fiber mats without incorporated icariin. Reprinted from work in [39]. Copyright 2017 with permission from Nature Publishing Group.

**Figure 11 polymers-11-02008-f011:**
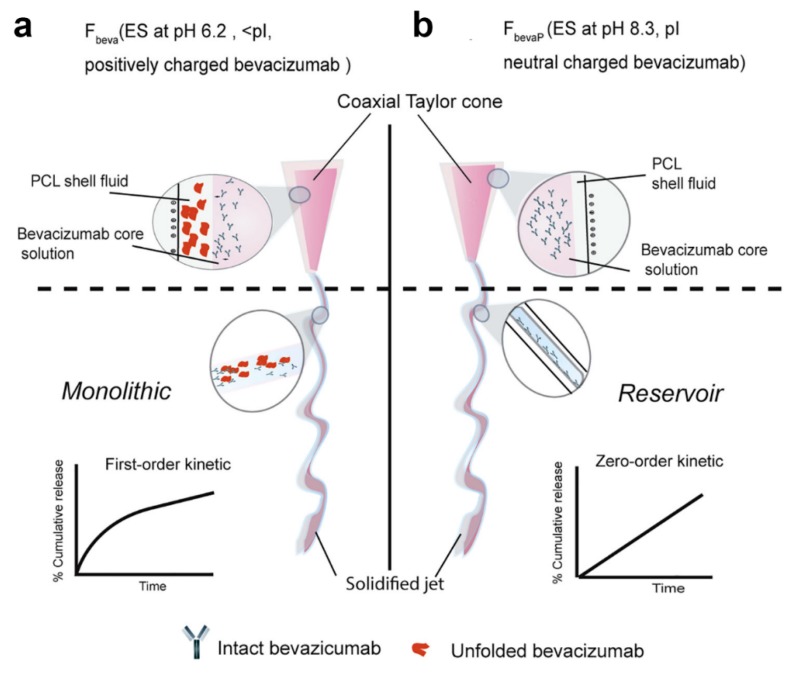
Schematic illustration describing the effect of different pH on drug distribution during electrospinning at (**a**) pH 6.2 and (**b**) pH 8.3. Reprinted from work in [209]. Copyright 2017 with permission from Elsevier.

**Figure 12 polymers-11-02008-f012:**
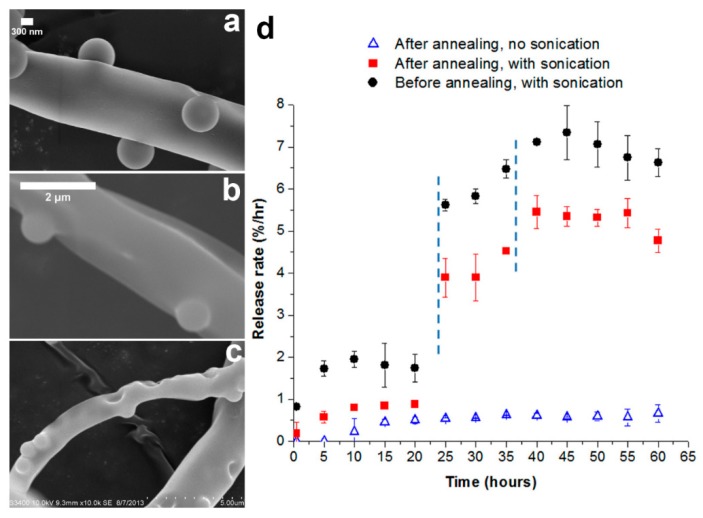
SEM images of PEO/PLA-silica nanoparticles (**a**) prior annealing, (**b**) post 30 minutes annealing, and (**c**) after 30 min ultrasound sonication. (**d**) Release profiles of rhodamine B which influenced by annealing and ultrasonication. Blue dashed line represents the point where 30 min sonication was applied. Reprinted from work in [106]. Copyright 2016 with permission from Elsevier.

**Figure 13 polymers-11-02008-f013:**
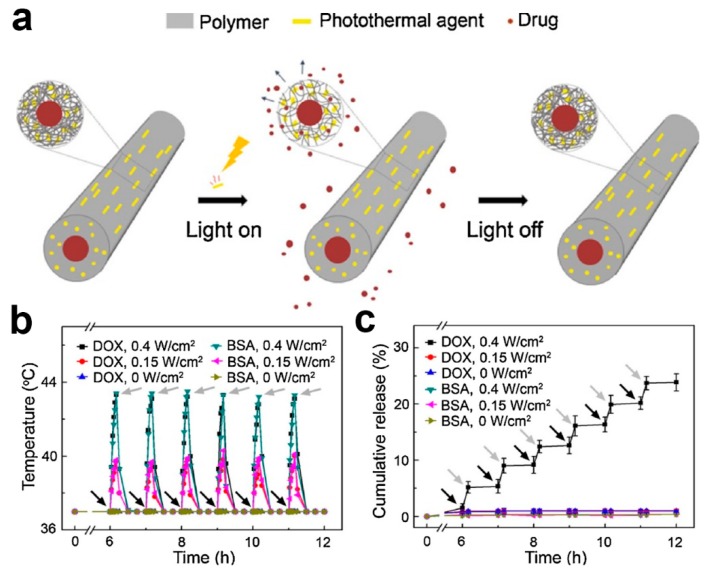
(**a**) Schematic diagram illustrating on-demand drug release triggered by NIR light. (**b**) Temperature curves of PLGA hollow fibers containing DOX and FITC-BSA as a function of time under on–off action of NIR light with varying power density. (**c**) Cumulative release profiles of DOX and FITC-BSA under on–off switching of NIR light. Black and grey arrow represents the point where NIR light was switched on and off, respectively. Reprinted from work in [118]. Copyright 2019 with permission from Elsevier.

**Table 1 polymers-11-02008-t001:** Selected combinations of core–shell fibers materials for various tissue engineering and drug delivery applications.

Core Material	Shell Material	Bioactive Molecules	Fabrication Technique	In vitro/in vivo Testing	Prospective Application	Ref.
PLA	PNIPAAM	Combreta-statin A4	Single electrospinning plus UV photopolymerization	Mouse fibroblast cells (L-929)	Biomaterial	[48]
Gelatin	Chitosan	na	Coaxial electrospinning	Human osteoblast cell line (MG-63)		[49]
na	PCL	Platelet lyophylisates	Emulsion centrifugal spinning	Human osteosarcoma cells (MG-63), murine 3T3 fibroblasts cells		[50]
PVP	PLGA	Naringin, metronidazole	Coaxial electrospinning	MC3T3-E1 cells	Guided tissue regeneration	[51]
PCL	Zein	Metronidazole	Coaxial electrospinning	L929 cells		[52]
PLGA/HA	Collagen	Amoxicillin	Coaxial electrospinning	HDF		[53]
PGS	PLA/PEO	na	Coaxial electrospinning	HUASMCs	Soft/hard tissue engineering	[54]
PCL	Collagen	na	Electrohydrodynamic plus bioprinting	Mouse preosteoblast (MC3T3-E1) cells		[55]
na	Collagen/chitosan/PLCL	Heparin	Coaxial electrospinning	PIECs	Vascular tissue engineering, vascular graft	[56]
PLLA/PEO	PLCL/PEO	na	Coaxial electrospinning	HUASMCs, HUVECs		[57]
na	PLCL/collagen	Heparin, Salvianolic acid B	Coaxial electrospinning	HUVECs/Male Sprague Dawley rats		[58]
na	PLGA	LBP	Coaxial electrospinning	Rat pheochromocytoma (PC12) cells	Nerve tissue engineering	[40]
na	PLGA, PDLLA	NGF, GDNF	Emulsion electrospinning	PC12 cells		[59]
PLLA	PGS	na	Single electrospinning plus phase separation	Hypothalamus A59 nerve cell		[60]
SF	PLA	NGF	Coaxial electrospinning	Rat PC12 cells	Neural tissue engineering	[29]
na	PDO/ collagen	Laminin	Magnetic-field assisted coaxial electrospinning	HT-22 mouse hippocampal neuronal cells		[61]
PLA	CA	Citalopram	Wet coaxial electrospinning	Rat Schwann cells/Male Wistar rats		[62]
PEG	PLGA	FGF-2	Coaxial electrospinning	PC12 cells/Male Wistar rats	Spinal cord tissue engineering	[23]
PCL	CMCh/ PVA	Zinc-curcumin complex	Coaxial electrospinning	Mouse fibroblast cells (L929), MG-63 human osteoblast cells	Bone tissue engineering	[63]
TSF/CaOH/H3PO4	TSF	na	Coaxial electrospinning	Human osteosarcoma MG-63 cells		[64]
PCL	PLA/HA	BMP-2	Coaxial electrospinning	hMSCs		[65]
na	SF/chitosan/nHAP	BMP-2	Coaxial electrospinning	BMMSCs/Female nude mice		[42]
na	PLGA/PCL	BMP-2	Coaxial electrospinning	rADSCs		[66]
na	SF/PLCL	Icariin	Coaxial electrospinning	BMMSCs/Male Sprague Dawley rats	Guided bone regeneration	[39]
na	SF/P(LLA-CL)	rhBMP-2, IGF-1	Coaxial electrospinning	BMMSCs		[67]
PCL/SF/PANI/CSA	PEGS-M	na	Single electrospinning plus UV irradiation	C2C12 mouse myoblasts	Skeletal muscle tissue engineering	[68]
CNTs	PELA	na	Coaxial electrospinning	Primary cardiomyocytes of neonatal rat	Cardiac tissue engineering	[33]
PCL	ShHL	na	Coaxial electrospinning	HUVECs, mouse fibroblast cells L929		[69]
CNTs	PELA	na	Coaxial electrospinning with micropatterned collector	CMs, ECs, CFs		[32]
PLA	Gelatin	na	Coaxial electrospinning	Rat chondrocyte, BMMSCs	Cartilage tissue engineering	[70]
na	P(LLA-CL)/collagen	Kartogenin	Coaxial electrospinning	BMMSCs	Tracheal cartilage regeneration	[11]
na	P(LLA-CL)/collagen	rhTGF-β3	Coaxial electrospinning	Human umbilical cord WMSCs		[12]
Zein prola-mine	Ethanol/DI water	GLSP	Coaxial electrospinning	Fibroblast L929 cells	Skin tissue engineering	[38]
PCL	PVA/ gelatin	Salvianolic acid B, bromelain	Coaxial electrospinning	Human epidermal keratinocytes, ECs/Female Wistar albino rats		[71]
na	SF/PEO	Dexametha-sone	Emulsion electrospinning	PHAECs		[72]
Poloxa-mer 188	PCL	Platelet lyophilisate	Needleless emulsion electrospin-ning, centrifugal force spinning	Murine XB2 cell line (keratinocytes), 3T3-A31 cell line (fibroblasts)	Dermal tissue engineering	[41]
PVP	PCL/ PVP	Sulfo-rhodamine B	Solution blow spinning	Human epidermal keratinocytes		[73]
PCL	PCL	na	Mechanical stretching	Human tenocytes/Male micropigs	Tendon tissue regeneration	[74]
PNIPA-AM	EC	Ketoprofen	Coaxial electrospinning	Mouse fibroblast cells (L929)	Advanced drug delivery	[75]
PVP/GO	PCL	Vancomycin hydrochloride	Coaxial electrospinning	L929 fibroblast cells		[76]
Hyalu-ronic acid	PCL	Ampicillin, Bay 11-7082, pirfenidone	Emulsion electrospinning plus electrospraying	Mouse embryonic fibroblasts (NIH3T3)/C57BL/6 mice	Drug eluting construct/stent	[37]
Gum traga-canth	PLGA	TCH	Coaxial electrospinning	HDF	Drug delivery-periodontal diseases	[77]
Chitosan	PCL	Ferulic acid, resveratrol	Coaxial electrospinning	Human epidermal keratinocytes/Female albino Wistar rats	Drug delivery-acute wounds	[78]
na	PLCL	EDTA, SC	Coaxial electrospinning	PIECs	Drug delivery-gallstone dissolution	[79]
PEO	Zein	Gallic acid	Coaxial electrospinning	Human gallbladder cancer cell lines (GB-d1 and NOZ)	Drug delivery-gallbladder cancer cells	[80]
PVA	SA/ PEO	Quercetin	Coaxial electrospinning	Colon cancer cells (Caco-2), mucosal cells (CCC-HIE-2)	Drug delivery-colon cancer	[81]
PES	PNIPAAM-co-Am	Curcumin	Single electrospinning plus coating (radical copolymerization)	Colon cancer cells HCT116		[82]
PVA	Gelatin/genipin	Doxorubicin	Coaxial electrospinning	4T1 cells (tumor cells), NIH 3T3 fibroblasts (normal cells)/4T1 tumor bearing nude mice	Cancer therapy	[83]
PCL	PCL/gelatin	Resveratrol, siRNA	Coaxial electrospinning	Erythroleukeia cell (K562)		[84]
PLGA/ PCL	Gelatin	Doxorubicin	Coaxial electrospinning	Mouse melanoma cell line (B16)/Female C57BL/6 mice	Skin cancer treatment	[35]
SF	PLCL/PEO	CTGF, FGF-2	Coaxial electrospinning	rMSCs	Mesenchymal stem cell trans-plantation	[85]
PVP	EC	Maraviroc	Coaxial electrospinning	TZM-bL cells	HIV prevention	[86]

Abbreviations: BMMSCs, bone marrow-derived mesenchymal stem cells; BMP-2, bone morphogenetic protein-2; CA, cellulose acetate; CFs, cardiac fibroblasts; CMCh, carboxymethyl chitosan; CMs, cardiomyocytes; CNTs, carbon nanotubes; CSA, camphorsulfonic acid; CTGF, connective tissue growth factor; DI, deionized; EC, ethyl cellulose; ECs, endothelial cells; EDTA, ethylene diamine tetraacetic acid; FGF-2, fibroblast growth factor-2; GDNF, glial cell line-derived neurotrophic factor; GLSP, ganoderma lucidum spore polysaccharide; GO, graphene oxide; HA, hydroxyapatite; HDF, human dermal fibroblasts; HIV, human immunodeficiency virus; hMSCs, human mesenchymal stem cells; HUASMCs, human umbilical artery smooth muscle cells; HUVECs, human umbilical vein endothelial cells; IGF-1, insulin growth factor-1; LBP, lycium barbarum polysaccharide; na, not applicable; NGF, nerve growth factor; nHAP, nanohydroxyapatite; PANI, polyaniline; PCL, poly(ɛ-caprolactone); PDLLA, poly(d,l-lactic acid); PDO, polydioxanone; PEG, poly(ethylene glycol); PEGS-M, poly(ethylene glycol)–*co*–poly(glycerol sebacate); PELA, poly(ethylene glycol)-poly(d,l-lactide); PEO, poly(ethylene oxide); PES, polyethersulfone; PGS, poly(glycerol sebacate); PHAECs, porcine hip artery endothelial cells; PIECs, porcine iliac endothelial cells; PLA, poly(lactic acid); PLCL, poly(lactide–*co*–ɛ-caprolactone); PLGA, poly(d,l-lactic–*co*–glycolic acid); PLLA, poly(l-lactic acid); P(LLA-CL), poly(l-lactide–*co*–caprolactone); PNIPAAM, poly(*N*-isopropylacrylamide); PVA, poly(vinyl alcohol); PVP, polyvinyl pyrrolidone; rADSCs, rat adipose-derived stem cells; rhBMP-2, recombinant human bone morphogenetic protein-2; rhTGF-β3, recombinant human transforming growth factor-β3; rMSCs, recombinant mesenchymal stem cells; SA, sodium alginate; SC, sodium cholate; SF, silk fibroin; ShHL, sulfated hydrolyzed halomonas levan; siRNA, small interfering ribonucleic acid; TCH, tetracycline hydrochloride; TSF, tussah silk fibroin; UV, ultraviolet; WMSCs, Wharton’s jelly mesenchymal stem cells.

**Table 2 polymers-11-02008-t002:** Summary of working principle, advantage, and limitation of remaining fabrication strategies of core–shell fibers.

Fabrication Technique	Working Principle	Advantage	Limitation	Ref.
Microfluidics	- Use special plate with slit channel where core flow channel is flanked by sheath flow channel - When laminar sheath flow flanks the core flow, core molecules were forced to align in flow direction - Aligned structure eventually frozen to form uniform core–shell fiber in gel phase	- Avoid use of high voltage	- Fiber size depends on channel diameter (currently at micro-size) - Low throughput	[134,135]
Solution blow spinning	- Require use of triaxial nozzle; for core and intermediate polymer, and compressed air (as shell fluid) - Airflow (10 psi) initiates solution spinning - The spinning caused solution to be drawn and formed fiber as a result of solvent evaporation	- Avoid electrostatic drive-force and conductive collector	- Large fiber diameter (∼1 µm) - Difficulty in producing aligned or patterned fiber	[73,133]
Coaxial airbrush	- Employing almost similar principle as solution blow spinning - Use concentric nozzle with three inlets; two for polymer solutions and one for compressed gas flow - High pressure gas (50–300 kPa) induces shearing at polymer solution/gas interface - Polymer solution deformed into conical shape and eventually yield core–shell fiber after solvent evaporated	- Avoid use of high voltage and conductive collector	- Relatively large average diameter of fiber (500 nm–1 µm) - Difficulty in future development of aligned and patterned fiber	[89]

**Table 3 polymers-11-02008-t003:** List of reported sensitive bioactive molecules and respective core–shell fibers systems employed to preserve them.

Bioactive Molecule		Limitation	Core System	Shell System	Ref.
Drug	Curcumin	Limited bioavailability due to poor absorption and rapid metabolism in body	Curcumin in absolute ethanol	PVA/chitosan in water/glacial acetic acid	[163]
	Resveratrol	Quickly metabolized and eliminated from body system (in form of sulfated and monoglucuronide derivatives)	Resveratrol/chitosan in acetic acid (90%)	PCL in DCM/ethanol	[78]
	Mycopheno-lic acid	Rapid decrease of concentration in vivo	Mycophenolic acid/PCL in TFE/DCM	PCL in TFE/DCM	[164]
	Tetracycline hydrochlo-ride	Vulnerable to oxidative degradation	Tetracycline hydrochloride/PVP in ethanol	PCL in acetic acid	[129]
	Berberine hydrochlo-ride	Low bioavailability post oral administration due to rapid decrease of plasma concentration	Berberine hydrochloride/ethylcellulose in acetone/ethanol	Glycerol monostearate in DCM/DMAc	[88]
Growth factor	VEGF	Short half-life (less than 1 h)	VEGF in BSA	P(LLA-CL)/collagen/elastin in HFIP	[90]
			Heparin/VEGF in distilled water	P(LLA-CL) in DCM	[119]
	PEDF	Short half-life in vivo and chemically unstable	PEDF/ PEG in DI water	PCL in DMF/chloroform	[165]
	NGF, GDNF	Potential denaturation and destabilization when in contact with organic solvent	GDNF in BSA, NGF in BSA	PLGA in chloroform, PDLLA in chloroform	[59]
Protein	Horseradish peroxidase	Potential loss of bioactivity due to conformation changes (caused by change of pH, temperature or UV light) and organic solvent interaction	Horseradish peroxidase in water	Eudragit® L100 in ethanol/DMF	[43]
Natural extract	Gallic acid	Unstable at alkaline pH, high temperature, and in presence of light or oxygen. Restricted absorption and quick excretion from body	Gallic acid/PEO in distilled water	Zein in ethanol/water	[80]

Abbreviations: BSA, bovine serum albumin; DCM, dichloromethane; DI, deionized; DMAc, dimethylacetamide; DMF, dimethylformamide; GDNF, glial cell line-derived neurotrophic factor; HFIP, hexafluoroisopropanol; NGF, nerve growth factor; PCL, poly(ɛ-caprolactone); PDLLA, poly(d,l-lactic acid); PEDF, pigmented epithelium-derived factor; PEG, poly(ethylene glycol); PEO, poly(ethylene oxide); PLGA, poly(d,l-lactic–*co*–glycolic acid); P(LLA-CL), poly(l-lactide–*co*–caprolactone); PVA, poly(vinyl alcohol); PVP, polyvinyl pyrrolidone; TFE, trifluoroethanol; UV, ultraviolet; VEGF, vascular endothelial growth factor.

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
