# Peer review of "Core–Shell Fibers: Design, Roles, and Controllable Release Strategies in Tissue Engineering and Drug Delivery"

_polymers, 2019, doi:10.3390/polym11122008_

Round 1

Reviewer 1 Report

This is a well-written and well-structured literature review and all figures are of very high quality. This manuscript summarizes all the previous reports of core-shell fibers for tissue engineering, drug delivery, and related therapies and is potentially useful to readers in genetic engineering and biopolymer science communities. One minor suggestion for this review manuscript: there is an important review paper by Špela Zupančič entitled "Core-shell nanofibers as drug delivery systems" that should have been discussed and referenced in the manuscript (Acta Pharmaceutica 2019, 69, 131–153). Similarly, electrospun fibers (Current Pharmaceutical Design, 2015, 21 1944–1959 and RSC Advances, 2019, 9, 25712-25729) have been investigated in detail and should therefore have been referenced, as well as several articles by the Huang group in Journal of Controlled Release 2014, 185, 12–21 and Biomaterials Science, 2018,6, 324-331. Given the above suggestions, this paper would be acceptable for the journal of Polymers after minor revision.

Reviewer 2 Report

In Table 1, the na abbreviation should be included, 

In line 314 reads: it is only requires, 

        should read: ..it only requires...

This a very well written and extensive review in a very good time for the state of the art in core-shell elctrospun fibers, and asess cleverly the advances and challenges for this area of knowledge.

Reviewer 3 Report

This is an exceptional and timely review by Muhammad and coworkers. They systematically summarized recent work of core-shell fibers for biomedical applications including engineering and drug delivery & release. The manuscript is very well written by the authors. The topic and the future direction as pointed by the authors are important for the field. Based on the high quality of this review, I will support publication of it in Polymers. Below are a few minor points.

1. Page 38, Lines 1038-1040 are related to the growing interest in smart and on-demand release. Those sentences should be corroborated by some recent literatures regarding on-demand release. For example, Rinaldi et al. ACS Applied Polymer Materials, 2019, 1, 211-220.
